# DiffAttack: Evasion Attacks Against Diffusion-Based Adversarial Purification

**Mintong Kang**
UIUC
mintong2@illinois.edu

**Dawn Song**
UC Berkeley
dawnsong@berkeley.edu

**Bo Li**
UIUC
lbo@illinois.edu

## Abstract

Diffusion-based purification defenses leverage diffusion models to remove crafted perturbations of adversarial examples and achieve state-of-the-art robustness. Recent studies show that even advanced attacks cannot break such defenses effectively, since the purification process induces an extremely deep computational graph which poses the potential problem of vanishing/exploding gradient, high memory cost, and unbounded randomness. In this paper, we propose an attack technique DiffAttack to perform effective and efficient attacks against diffusion-based purification defenses, including both DDPM and score-based approaches. In particular, we propose a deviated-reconstruction loss at intermediate diffusion steps to induce inaccurate density gradient estimation to tackle the problem of vanishing/exploding gradients. We also provide a segment-wise forwarding-backwarding algorithm, which leads to memory-efficient gradient backpropagation. We validate the attack effectiveness of DiffAttack compared with existing adaptive attacks on CIFAR-10 and ImageNet. We show that DiffAttack decreases the robust accuracy of models compared with SOTA attacks by over 20% on CIFAR-10 under $\ell_\infty$ attack ($\epsilon = 8/255$), and over 10% on ImageNet under $\ell_\infty$ attack ($\epsilon = 4/255$). We conduct a series of ablations studies, and we find 1) DiffAttack with the deviated-reconstruction loss added over uniformly sampled time steps is more effective than that added over only initial/final steps, and 2) diffusion-based purification with a moderate diffusion length is more robust under DiffAttack.

## 1 Introduction

Since deep neural networks (DNNs) are found vulnerable to adversarial perturbations [52, 20], improving the robustness of neural networks against such crafted perturbations has become important, especially in safety-critical applications [18, 5, 54]. In recent years, many defenses have been proposed, but they are attacked again by more advanced adaptive attacks [7, 30, 11, 12]. One recent line of defense (*diffusion-based purification*) leverages diffusion models to purify the input images and achieves the state-of-the-art robustness. Based on the type of diffusion models the defense utilizes, diffusion-based purification can be categorized into *score-based purification* [34] which uses the score-based diffusion model [49] and *DDPM-based purification* [4, 62, 57, 51, 55, 56] which uses the denoising diffusion probabilistic model (DDPM) [25]. Recent studies show that even the most advanced attacks [12, 34] cannot break these defenses due to the challenges of vanishing/exploding gradient, high memory cost, and large randomness. In this paper, we aim to explore the vulnerabilities of such diffusion-based purification defenses, and *design a more effective and efficient adaptive attack against diffusion-based purification*, which will help to better understand the properties of diffusion process and motivate future defenses.

In particular, the diffusion-based purification defenses utilize diffusion models to first diffuse the adversarial examples with Gaussian noises and then perform sampling to remove the noises. In this way, the hope is that the crafted adversarial perturbations can also be removed since the training

37th Conference on Neural Information Processing Systems (NeurIPS 2023).

distribution of diffusion models is clean [49, 25]. The diffusion length (i.e., the total diffusion time steps) is usually large, and at each time step, the deep neural network is used to estimate the gradient of the data distribution. This results in an extremely deep computational graph that poses great challenges of attacking it: *vanishing/exploding gradients*, *unavailable memory cost*, and *large randomness*. To tackle these challenges, we propose a deviated-reconstruction loss and a segment-wise forwarding-backwarding algorithm and integrate them as an effective and efficient attack technique *DiffAttack*.

Essentially, our **deviated-reconstruction loss** pushes the reconstructed samples away from the diffused samples at corresponding time steps. It is added at multiple intermediate time steps to relieve the problem of vanishing/exploding gradients. We also theoretically analyze the connection between it and the score-matching loss [26], and we prove that maximizing the deviated-reconstruction loss induces inaccurate estimation of the density gradient of the data distribution, leading to a higher chance of attacks. To overcome the problem of large memory cost, we propose a **segment-wise forwarding-backwarding** algorithm to backpropagate the gradients through a long path. Concretely, we first do a forward pass and store intermediate samples, and then iteratively simulate the forward pass of a segment and backward the gradient following the chain rule. Ignoring the memory cost induced by storing samples (small compared with the computational graph), our approach achieves $\mathcal{O}(1)$ memory cost.

Finally, we integrate the deviated-reconstruction loss and segment-wise forwarding-backwarding algorithm into DiffAttack, and empirically validate its effectiveness on CIFAR-10 and ImageNet. We find that (1) DiffAttack outperforms existing attack methods [34, 60, 53, 1, 2] by a large margin for both the score-based purification and DDPM-based purification defenses, especially under large perturbation radii; (2) the memory cost of our efficient segment-wise forwarding-backwarding algorithm does not scale up with the diffusion length and saves more than 10x memory cost compared with the baseline [4]; (3) a moderate diffusion length benefits the robustness of the diffusion-based purification since longer length will hurt the benign accuracy while shorter length makes it easier to be attacked; (4) attacks with the deviated-reconstruction loss added over uniformly sampled time steps outperform that added over only initial/final time steps. The effectiveness of DiffAttack and interesting findings will motivate us to better understand and rethink the robustness of diffusion-based purification defenses.

We summarize the main *technical contributions* as follows:

- We propose DiffAttack, a strong evasion attack against the diffusion-based adversarial purification defenses, including score-based and DDPM-based purification.
- We propose a deviated-reconstruction loss to tackle the problem of vanishing/exploding gradient, and theoretically analyze its connection with data density estimation.
- We propose a segment-wise forwarding-backwarding algorithm to tackle the high memory cost challenge, and we are the *first* to adaptively attack the DDPM-based purification defense, which is hard to attack due to the high memory cost.
- We empirically demonstrate that DiffAttack outperforms existing attacks by a large margin on CIFAR-10 and ImageNet. Particularly, DiffAttack decreases the model robust accuracy by over 20% for $\ell_\infty$ attack ($\epsilon = 8/255$) on CIFAR-10, and over 10% on ImageNet under $\ell_\infty$ attack ($\epsilon = 4/255$).
- We conduct a series of ablation studies and show that (1) a moderate diffusion length benefits the model robustness, and (2) attacks with the deviated-reconstruction loss added over uniformly sampled time steps outperform that added over only initial/final time steps.

## 2   Preliminary

There are two types of diffusion-based purification defenses, **DDPM-based purification**, and **score-based purification**, which leverage *DDPM* [46, 25] and *score-based diffusion model* [49] to purify the adversarial examples, respectively. Next, we will introduce the basic concepts of DDPM and score-based diffusion models.

Denote the diffusion process indexed by time step $t$ with the *diffusion length* $T$ by $\{\mathbf{x}_t\}_{t=0}^T$. DDPM constructs a discrete Markov chain $\{\mathbf{x}_t\}_{t=0}^T$ with discrete time variables $t$ following $p(\mathbf{x}_t|\mathbf{x}_{t-1}) = \mathcal{N}(\mathbf{x}_t; \sqrt{1-\beta_t}\mathbf{x}_{t-1}, \beta_t\mathbf{I})$ where $\beta_t$ is a sequence of positive noise scales (e.g., linear scheduling, cosine scheduling [33]). Considering $\alpha_t := 1 - \beta_t$, $\bar{\alpha}_t := \Pi_{s=1}^t \alpha_s$, and

$\sigma_t = \sqrt{\beta_t(1 - \bar{\alpha}_{t-1})/(1 - \bar{\alpha}_t)}$, the reverse process (i.e., sampling process) can be formulated as:

$$\mathbf{x}_{t-1} = \frac{1}{\sqrt{\alpha_t}} \left( \mathbf{x}_t - \frac{1 - \alpha_t}{\sqrt{1 - \bar{\alpha}_t}} \boldsymbol{\epsilon}_\theta(\mathbf{x}_t, t) \right) + \sigma_t \mathbf{z} \tag{1}$$

where $\mathbf{z}$ is drawn from $\mathcal{N}(\mathbf{0}, \mathbf{I})$. $\boldsymbol{\epsilon}_\theta$ parameterized with $\theta$ is the model to approximate the perturbation $\boldsymbol{\epsilon}$ in the diffusion process and is trained via the *density gradient loss* $\mathcal{L}_d$:

$$\mathcal{L}_d = \mathbb{E}_{t,\boldsymbol{\epsilon}} \left[ \frac{\beta_t^2}{2\sigma_t^2 \alpha_t (1 - \bar{\alpha}_t)} \| \boldsymbol{\epsilon} - \boldsymbol{\epsilon}_\theta(\sqrt{\bar{\alpha}_t}\mathbf{x}_0 + \sqrt{1 - \bar{\alpha}_t}\boldsymbol{\epsilon}, t) \|_2^2 \right] \tag{2}$$

where $\boldsymbol{\epsilon}$ is drawn from $\mathcal{N}(\mathbf{0}, \mathbf{I})$ and $t$ is uniformly sampled from $[T] := \{1, 2, ..., T\}$.

Score-based diffusion model formulates diffusion models with stochastic differential equations (SDE). The diffusion process $\{\mathbf{x}_t\}_{t=0}^T$ is indexed by a continuous time variable $t \in [0, 1]$. The diffusion process can be formulated as:

$$d\mathbf{x} = f(\mathbf{x}, t)dt + g(t)d\mathbf{w} \tag{3}$$

where $f(\mathbf{x}, t) : \mathbb{R}^n \mapsto \mathbb{R}^n$ is the drift coefficient characterizing the shift of the distribution, $g(t)$ is the diffusion coefficient controlling the noise scales, and $\mathbf{w}$ is the standard Wiener process. The reverse process is characterized via the reverse time SDE of Equation (3):

$$d\mathbf{x} = [f(\mathbf{x}, t) - g(t)^2 \nabla_\mathbf{x} \log p_t(\mathbf{x})]dt + g(t)d\mathbf{w} \tag{4}$$

where $\nabla_\mathbf{x} \log p_t(\mathbf{x})$ is the time-dependent score function that can be approximated with neural networks $\mathbf{s}_\theta$ parameterized with $\theta$, which is trained via the score matching loss $\mathcal{L}_s$ [26, 47]:

$$\mathcal{L}_s = \mathbb{E}_t \left[ \lambda(t) \mathbb{E}_{\mathbf{x}_t | \mathbf{x}_0} \| \mathbf{s}_\theta(\mathbf{x}_t, t) - \nabla_{\mathbf{x}_t} \log(p(\mathbf{x}_t | \mathbf{x}_0)) \|_2^2 \right] \tag{5}$$

where $\lambda : [0, 1] \to \mathbb{R}$ is a weighting function and $t$ is uniformly sampled over $[0, 1]$.

## 3   DiffAttack

### 3.1   Evasion attacks against diffusion-based purification

A class of defenses leverages generative models for adversarial purification [43, 48, 45, 60]. The adversarial images are transformed into latent representations, and then the purified images are sampled starting from the latent space using the generative models. The process is expected to remove the crafted perturbations since the training distribution of generative models is assumed to be clean. With diffusion models showing the power of image generation recently [15, 39], diffusion-based adversarial purification has achieved SOTA defense performance [34, 4].

We first formulate the problem of evasion attacks against diffusion-based purification defenses. Suppose that the process of diffusion-based purification, including the diffusion and reverse process, is denoted by $P : \mathbb{R}^n \mapsto \mathbb{R}^n$ where $n$ is the dimension of the input $\mathbf{x}_0$, and the classifier is denoted by $F : \mathbb{R}^n \mapsto [K]$ where $K$ is the number of classes. Given an input pair $(\mathbf{x}_0, y)$, the adversarial example $\tilde{\mathbf{x}}_0$ satisfies:

$$\underset{i \in [K]}{\arg\max} \, F_i(P(\tilde{\mathbf{x}}_0)) \neq y \quad s.t. \; d(\mathbf{x}_0, \tilde{\mathbf{x}}_0) \leq \delta_{max} \tag{6}$$

where $F_i(\cdot)$ is the $i$-th element of the output, $d : \mathbb{R}^n \times \mathbb{R}^n \mapsto \mathbb{R}$ is the distance function in the input space, and $\delta_{max}$ is the perturbation budget.

Since directly searching for the adversarial instance $\tilde{\mathbf{x}}_0$ based on Equation (6) is challenging, we often use a surrogate loss $\mathcal{L}$ to solve an optimization problem:

$$\underset{\tilde{\mathbf{x}}_0}{\max} \, \mathcal{L}(F(P(\tilde{\mathbf{x}}_0)), y) \quad s.t. \; d(\mathbf{x}_0, \tilde{\mathbf{x}}_0) \leq \delta_{max} \tag{7}$$

where $P(\cdot)$ is the purification process with DDPM (Equation (1)) or score-based diffusion (Equations (3) and (4)), and the surrogate loss $\mathcal{L}$ is often selected as the classification-guided loss, such as CW loss [7], Cross-Entropy loss and difference of logits ratio (DLR) loss [12]. Existing adaptive attack methods such as PGD [30] and APGD attack [12] approximately solve the optimization problem in Equation (7) via computing the gradients of loss $\mathcal{L}$ with respect to the decision variable $\tilde{\mathbf{x}}_0$ and iteratively updating $\tilde{\mathbf{x}}_0$ with the gradients.

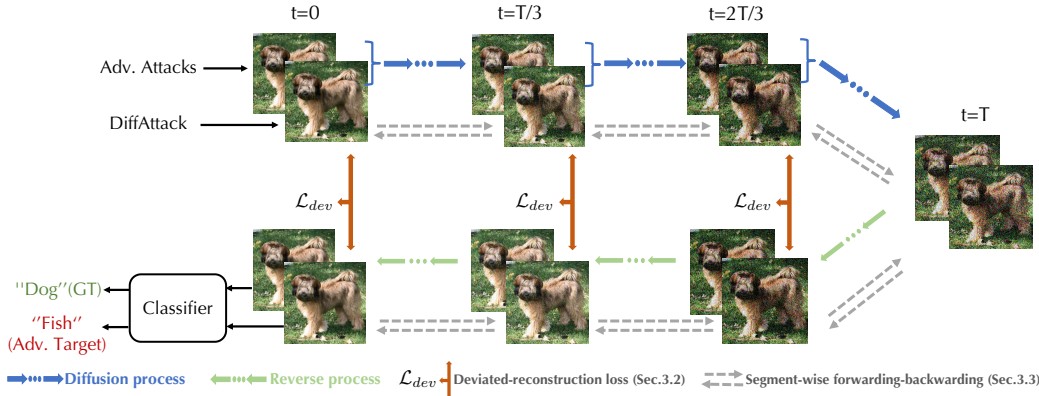

Figure 1: DiffAttack against diffusion-based adversarial purification defenses. DiffAttack features the *deviated-reconstruction loss* that addresses vanishing/exploding gradients and the *segment-wise forwarding-backwarding algorithm* that leads to memory-efficient gradient backpropagation.

However, we observe that the gradient computation for the diffusion-based purification process is challenging for *three* reasons: 1) the long sampling process of the diffusion model induces an extremely deep computational graph which poses the problem of vanishing/exploding gradient [2], 2) the deep computational graph impedes gradient backpropagation, which requires high memory cost [60, 4], and 3) the diffusion and sampling process introduces large randomness which makes the calculated gradients unstable and noisy.

To address these challenges, we propose a deviated-reconstruction loss (in Section 3.2) and a segment-wise forwarding-backwarding algorithm (in Section 3.3) and design an effective algorithm DiffAttack by integrating them into the attack technique (in Section 3.4).

## 3.2 Deviated-reconstruction loss

In general, the surrogate loss $\mathcal{L}$ in Equation (7) is selected as the classification-guided loss, such as CW loss, Cross-Entropy loss, or DLR loss. However, these losses can only be imposed at the classification layer, and induce the problem of vanishing/exploding gradients [2] due to the long diffusion length. Specifically, the diffusion purification process induces an extremely deep graph. For example, DiffPure applies hundreds of iterations of sampling and uses deep UNet with tens of layers as score estimators. Thus, the computational graph consists of thousands of layers, which could cause the problem of gradient vanishing/exploding. Similar gradient problems are also mentioned with generic score-based generative purification (Section 4, 5.1 in [60]). Backward path differentiable approximation (BPDA) attack [2] is usually adopted to overcome such problems, but the surrogate model of the complicated sampling process is hard to find, and a simple identity mapping function is demonstrated to be ineffective in the case [34, 4, 60].

To overcome the problem of exploding/vanishing gradients, we attempt to impose intermediate guidance during the attack. It is possible to build a set of classifiers on the intermediate samples in the reverse process and use the weighted average of the classification-guided loss at multiple layers as the surrogate loss $\mathcal{L}$. However, we observe that the intermediate samples are noisy, and thus using classifier $F$ that is trained on clean data cannot provide effective gradients. One solution is to train a set of classifiers with different noise scales and apply them to intermediate samples to impose classification-guided loss, but the training is too expensive considering the large diffusion length and variant noise scales at different time steps. Thus, we propose a deviated-reconstruction loss to address the challenge via imposing discrepancy for samples between the diffusion and reverse processes adversarially to provide effective loss at intermediate time steps.

Concretely, since a sequence of samples is generated in the diffusion and reverse processes, effective loss imposed on them would relieve the problem of vanishing/exploding gradient and benefit the optimization. More formally, let $\mathbf{x}_t$, $\mathbf{x}_t'$ be the samples at time step $t$ in the diffusion process and the reverse process, respectively. Formally, we maximize the deviated-reconstruction loss $\mathcal{L}_{dev}$

formulated as follows:

$$\max \mathcal{L}_{dev} = \mathbb{E}_t[\alpha(t)\mathbb{E}_{\mathbf{x}_t,\mathbf{x}'_t|\mathbf{x}_0}d(\mathbf{x}_t,\mathbf{x}'_t)] \tag{8}$$

where $\alpha(\cdot)$ is time-dependent weight coefficients and $d(\mathbf{x}_t, \mathbf{x}'_t)$ is the distance between noisy image $\mathbf{x}_t$ in the diffusion process and corresponding sampled image $\mathbf{x}'_t$ in the reverse process. The expectation over $t$ is approximated by taking the average of results at uniformly sampled time steps in $[0, T]$, and the loss at shallow layers in the computational graph (i.e., large time step $t$) helps relieve the problem of vanishing/exploding gradient. The conditional expectation over $\mathbf{x}_t, \mathbf{x}'_t$ given $\mathbf{x}_0$ is approximated by purifying $\mathbf{x}_0$ multiple times and taking the average of the loss.

Intuitively, the deviated-reconstruction loss in Equation (8) pushes the reconstructed sample $\mathbf{x}'_t$ in the reverse process away from the sample $\mathbf{x}_t$ at the corresponding time step in the diffusion process, and finally induces an inaccurate reconstruction of the clean image. Letting $q_t(\mathbf{x})$ and $q'_t(\mathbf{x})$ be the distribution of $\mathbf{x}_t$ and $\mathbf{x}'_t$, we can theoretically prove that the distribution distance between $q_t(\mathbf{x})$ and $q'_t(\mathbf{x})$ positively correlates with the score-matching loss of the score-based diffusion or the density gradient loss of the DDPM. In other words, maximizing the deviated-reconstruction loss in Equation (8) induces inaccurate data density estimation, which results in the discrepancy between the sampled distribution and the clean training distribution.

**Theorem 1.** *Consider adversarial sample $\tilde{\mathbf{x}}_0 := \mathbf{x}_0 + \delta$, where $\mathbf{x}_0$ is the clean example and $\delta$ is the perturbation. $p_t(\mathbf{x}), p'_t(\mathbf{x}), q_t(\mathbf{x}), q'_t(\mathbf{x})$ are the distribution of $\mathbf{x}_t, \mathbf{x}'_t, \tilde{\mathbf{x}}_t, \tilde{\mathbf{x}}'_t$ where $\mathbf{x}'_t$ represents the reconstruction of $\mathbf{x}_t$ in the reverse process. $D_{TV}(\cdot, \cdot)$ measures the total variation distance. Given a VP-SDE parameterized by $\beta(\cdot)$ and the score-based model $\mathbf{s}_\theta$ with mild assumptions that $\|\nabla_{\mathbf{x}} \log p_t(\mathbf{x}) - \mathbf{s}_\theta(\mathbf{x}, t)\|_2^2 \leq L_u$, $D_{TV}(p_t, p'_t) \leq \epsilon_{re}$, and a bounded score function by $M$ (specified in Appendix C.1), we have:*

$$D_{TV}(q_t, q'_t) \leq \frac{1}{2}\sqrt{\mathbb{E}_{t,\mathbf{x}|\mathbf{x}_0}\|\mathbf{s}_\theta(\mathbf{x}, t) - \nabla_{\mathbf{x}} \log q'_t(\mathbf{x})\|_2^2 + C_1} + \sqrt{2 - 2\exp\{-C_2\|\delta\|_2^2\}} + \epsilon_{re} \tag{9}$$

$C_1 = (L_u + 8M^2)\int_t^T \beta(t)dt$, $C_2 = (8(1 - \Pi_{s=1}^t(1 - \beta_s)))^{-1}$.

*Proof sketch.* We first use the triangular inequality to upper bound $D_{TV}(q_t, q'_t)$ with $D_{TV}(q_t, p_t) + D_{TV}(p_t, p'_t) + D_{TV}(p'_t, q'_t)$. $D_{TV}(q_t, p_t)$ can be upper bounded by a function of the Hellinger distance $H(q_t, p_t)$, which can be calculated explicitly. $D_{TV}(p_t, p'_t)$ can be upper bounded by the reconstruction error $\epsilon_{re}$ by assumption. To upper bound $D_{TV}(p'_t, q'_t)$, we can leverage Pinker's inequality to alternatively upper bound the KL-divergence between $p'_t$ and $q'_t$ which can be derived by using the Fokker-Planck equation [44] in the reverse SDE.

*Remark.* A large deviated-reconstruction loss can indicate a large total variation distance $D_{TV}(q_t, q'_t)$, which is the lower bound of a function with respect to the score-matching loss $\mathbb{E}_{t,\mathbf{x}}\|\mathbf{s}_\theta(\mathbf{x}, t) - \nabla_{\mathbf{x}} \log q'_t(\mathbf{x})\|_2^2$ (in RHS of Equation (9)). Therefore, we show that maximizing the deviated-reconstruction loss implicitly maximizes the score-matching loss, and thus induces inaccurate data density estimation to perform an effective attack. The connection of deviated-reconstruction loss and the density gradient loss for DDPM is provided in Thm. 3 in Appendix C.2.

## 3.3 Segment-wise forwarding-backwarding algorithm

Adaptive attacks against diffusion-based purification require gradient backpropagation through the forwarding path. For diffusion-based purification, the memory cost scales linearly with the diffusion length $T$ and is not feasible in a realistic application. Therefore, existing defenses either use a surrogate model for gradient approximation [55, 56, 60, 45] or consider adaptive attacks only for a small diffusion length [4], but the approximation can induce error and downgrade the attack performance a lot. Recently, DiffPure [34] leverages the adjoint method [28] to backpropagate the gradient of SDE within reasonable memory cost and enables adaptive attacks against score-based purification. However, it cannot be applied to a discrete process, and the memory-efficient gradient backpropagation algorithm is unexplored for DDPM. Another line of research [9, 8, 19] proposes the technique of gradient checkpointing to perform gradient backpropagation with memory efficiency. Fewer activations are stored during forwarding passes, and the local computation graph is constructed via recomputation. However, we are the first to apply the memory-efficient backpropagation technique to attack diffusion purification defenses and resolve the problem of memory cost during attacks, which is realized as a challenging problem by prior attacks against purification defenses [34, 60]. Concretely, we propose a segment-wise forwarding-backwarding algorithm, which leads to memory-efficient gradient computation of the attack loss with respect to the adversarial examples.

We first feed the input $\mathbf{x}_0$ to the diffusion-based purification process and store the intermediate samples $\mathbf{x}_1, \mathbf{x}_2, ..., \mathbf{x}_T$ in the diffusion process and $\mathbf{x}'_T, \mathbf{x}'_{T-1}, ..., \mathbf{x}'_0$ in the reverse process sequentially. For ease of notation, we have $\mathbf{x}_{t+1} = f_d(\mathbf{x}_t)$ and $\mathbf{x}'_t = f_r(\mathbf{x}'_{t+1})$ for $t \in [0, T-1]$. Then we can backpropagate the gradient iteratively following:

$$\frac{\partial \mathcal{L}}{\partial \mathbf{x}'_{t+1}} = \frac{\partial \mathcal{L}}{\partial \mathbf{x}'_t} \frac{\partial \mathbf{x}'_t}{\partial \mathbf{x}'_{t+1}} = \frac{\partial \mathcal{L}}{\partial \mathbf{x}'_t} \frac{\partial f_r(\mathbf{x}'_{t+1})}{\partial \mathbf{x}'_{t+1}} \tag{10}$$

At each time step $t$ in the reverse process, we only need to store the gradient $\partial \mathcal{L} / \partial \mathbf{x}'_t$, the intermediate sample $\mathbf{x}'_{t+1}$ and the model $f_r$ to construct the computational graph. When we backpropagate the gradients at the next time step $t + 1$, the computational graph at time step $t$ will no longer be reused, and thus, we can release the memory of the graph at time step $t$. Therefore, we only have *one segment of the computational graph* used for gradient backpropagation in the memory at each time step. We can similarly backpropagate the gradients in the diffusion process. Ignoring the memory cost of storing intermediate samples (usually small compared to the memory cost of computational graphs), the memory cost of our segment-wise forwarding-backwarding algorithm is $\mathcal{O}(1)$ (validated in Figure 3).

We summarize the detailed procedures in Algorithm 1 in Appendix B. It can be applied to gradient backpropagation through any discrete Markov process with a long path. Basically, we *1) perform the forward pass and store the intermediate samples, 2) allocate the memory of one segment of the computational graph in the memory and simulate the forwarding pass of the segment with intermediate samples, 3) backpropagate the gradients through the segment and release the memory of the segment, and 4) go to step 2 and consider the next segment until termination.*

### 3.4 DiffAttack Technique

Currently, AutoAttack [12] holds the state-of-the-art attack algorithm, but it fails to attack the diffusion-based purification defenses due to the challenge of *vanishing/exploding gradient*, *memory cost* and *large randomness*. To specifically tackle the challenges, we integrate the deviated-reconstruction loss (in Section 3.2) and the segment-wise forwarding-backwarding algorithm (in Section 3.3) as an attack technique *DiffAttack* against diffusion-based purification, including the score-based and DDPM-based purification defenses. The pictorial illustration of DiffAttack is provided in Figure 1.

Concretely, we maximize the surrogate loss $\mathcal{L}$ as the optimization objective in Equation (7):

$$\max \mathcal{L} = \mathcal{L}_{cls} + \lambda \mathcal{L}_{dev} \tag{11}$$

where $\mathcal{L}_{cls}$ is the CE loss or DLR loss, $\mathcal{L}_{dev}$ is the deviated-reconstruction loss formulated in Equation (8), and $\lambda$ is the weight coefficient. During the optimization, we use the segment-wise forwarding-backwarding algorithm for memory-efficient gradient backpropagation. Note that $\mathcal{L}_{dev}$ suffers less from the gradient problem compared with $\mathcal{L}_{cls}$, and thus the objective of $\mathcal{L}_{dev}$ can be optimized more precisely and stably, but it does not resolve the gradient problem of $\mathcal{L}_{cls}$. On the other hand, the optimization of $\mathcal{L}_{dev}$ benefits the optimization of $\mathcal{L}_{cls}$ in the sense that $\mathcal{L}_{dev}$ can induce a deviated reconstruction of the image with a larger probability of misclassification. $\lambda$ controls the balance of the two objectives. A small $\lambda$ can weaken the deviated-reconstruction object and make the attack suffer more from the vanishing/exploded gradient problem, while a large $\lambda$ can downplay the guidance of the classification loss and confuse the direction towards the decision boundary of the classifier.

*Attack against randomized diffusion-based purification.* DiffAttack tackles the randomness problem from two perspectives: 1) sampling the diffused and reconstructed samples across different time steps multiple times as in Equation (8) (similar to EOT [3]), and 2) optimizing perturbations for all samples including misclassified ones in all steps. Perspective 1) provides a more accurate estimation of gradients against sample variance of the diffusion process. Perspective 2) ensures a more effective and stable attack optimization since the correctness of classification is of high variance over different steps in the diffusion purification setting. Formally, the classification result of a sample can be viewed as a Bernoulli distribution (i.e., correct or false). We should reduce the success rate of the Bernoulli distribution of sample classification by optimizing them with a larger attack loss, which would lead to lower robust accuracy. In other words, one observation of failure in classification does not indicate that the sample has a low success rate statistically, and thus, perspective 2) helps

Table 1: Attack performance (Rob-Acc (%)) of DiffAttack and AdjAttack [34] against score-based purification on CIFAR-10.

| Models | T | Cl-Acc | $\ell_p$ Attack | $\epsilon$ | Method | Rob-Acc | Diff. |
|---|---|---|---|---|---|---|---|
| WideResNet-28-10 | 0.1 | 89.02 | $\ell_\infty$ | 8/255 | AdjAttack
DiffAttack | 70.64
**46.88** | **-23.76** |
| | | | | 4/255 | AdjAttack
DiffAttack | 82.81
**71.88** | **-10.93** |
| | 0.075 | 91.03 | $\ell_2$ | 0.5 | AdjAttack
DiffAttack | 78.58
**64.06** | **-14.52** |
| WideResNet-70-16 | 0.1 | 90.07 | $\ell_\infty$ | 8/255 | AdjAttack
DiffAttack | 71.29
**45.31** | **-25.98** |
| | | | | 4/255 | AdjAttack
DiffAttack | 81.25
**75.00** | **-6.25** |
| | 0.075 | 92.68 | $\ell_2$ | 0.5 | AdjAttack
DiffAttack | 80.60
**70.31** | **-10.29** |

to continue optimizing the perturbations towards a lower success rate (i.e., away from the decision boundary). We provide the pseudo-codes of DiffAttack in Algorithm 2 in Appendix D.1.

## 4 Experimental Results

In this section, we evaluate DiffAttack from various perspectives empirically. As a summary, we find that 1) DiffAttack significantly outperforms other SOTA attack methods against diffusion-based defenses on both the score-based purification and DDPM-based purification models, especially under large perturbation radii (Section 4.2 and Section 4.3); 2) DiffAttack outperforms other strong attack methods such as the black-box attack and adaptive attacks against other adversarial purification defenses (Section 4.4); 3) a moderate diffusion length $T$ benefits the model robustness, since too long/short diffusion length would hurt the robustness (Section 4.5); 4) our proposed segment-wise forwarding-backwarding algorithm achieves $\mathcal{O}(1)$-memory cost and outperforms other baselines by a large margin (Section 4.6); and 5) attacks with the deviated-reconstruction loss added over uniformly sampled time steps outperform that added over only initial/final time steps (Section 4.7).

### 4.1 Experiment Setting

**Dataset & model.** We validate DiffAttack on CIFAR-10 [27] and ImageNet [13]. We consider different network architectures for classification. Particularly, WideResNet-28-10 and WideResNet-70-16 [61] are used on CIFAR-10, and ResNet-50 [23], WideResNet-50-2 (WRN-50-2), and ViT (DeiT-S) [16] are used on ImageNet. We use a pretrained score-based diffusion model [49] and DDPM [25] to purify images following [34, 4].

**Evaluation metric.** The performance of attacks is evaluated using the *robust accuracy* (Rob-Acc), which measures the ratio of correctly classified instances over the total number of test data under certain perturbation constraints. Following the literature [12], we consider both $\ell_\infty$ and $\ell_2$ attacks under multiple perturbation constraints $\epsilon$. We also report the clean accuracy (Cl-Acc) for different approaches.

**Baselines.** To demonstrate the effectiveness of DiffAttack, we compare it with 1) SOTA attacks against score-based diffusion *adjoint attack* (AdjAttack) [34], 2) SOTA attack against DDPM-based diffusion *Diff-BPDA attack* [4], 3) SOTA black-box attack *SPSA* [53] and *square attack* [1], and 4) specific attack against EBM-based purification *joint attack* [60]. We defer more explanations of baselines and experiment details to Appendix D.2. The codes are publicly available at https://github.com/kangmintong/DiffAttack.

Table 3: Attack performance (Rob-Acc (%)) of DiffAttack and Diff-BPDA [4] against DDPM-based purification on CIFAR-10.

| Architecture | T | Cl-Acc | $\ell_p$ Attack | $\epsilon$ | Method | Rob-Acc | Diff. |
|---|---|---|---|---|---|---|---|
| WideResNet-28-10 | 100 | 87.50 | $\ell_\infty$ | 8/255 | Diff-BPDA
DiffAttack | 75.00
**54.69** | **-20.31** |
| | | | | 4/255 | Diff-BPDA
DiffAttack | 76.56
**63.28** | **-13.28** |
| | 75 | 90.62 | $\ell_2$ | 0.5 | Diff-BPDA
DiffAttack | 76.56
**67.97** | **-8.59** |
| WideResNet-70-16 | 100 | 91.21 | $\ell_\infty$ | 8/255 | Diff-BPDA
DiffAttack | 74.22
**59.38** | **-14.84** |
| | | | | 4/255 | Diff-BPDA
DiffAttack | 75.78
**67.19** | **-8.59** |
| | 75 | 92.19 | $\ell_2$ | 0.5 | Diff-BPDA
DiffAttack | 81.25
**71.88** | **-9.37** |

## 4.2 Attack against score-based purification

DiffPure [34] presents the state-of-the-art adversarial purification performance using the score-based diffusion models [49]. It proposes a strong adaptive attack (AdjAttack) which uses the adjoint method [28] to efficiently backpropagate the gradients through reverse SDE. Therefore, we select AdjAttack as the strong baseline and compare DiffAttack with it. The results on CIFAR-10 in Table 1 show that DiffAttack achieves much lower robust accuracy compared with AdjAttack under different types of attacks ($\ell_\infty$ and $\ell_2$ attack) with multiple perturbation constraints $\epsilon$. Concretely, DiffAttack decreases the robust accuracy

Table 2: Attack performance of DiffAttack and AdjAttack [34] against score-based adversarial purification with diffusion length $T = 0.015$ on ImageNet under $\ell_\infty$ attack ($\epsilon = 4/255$).

| Models | Cl-Acc | Method | Rob-Acc | Diff. |
|---|---|---|---|---|
| ResNet-50 | 67.79 | AdjAttack
DiffAttack | 40.93
**28.13** | **-12.80** |
| WRN-50-2 | 71.16 | AdjAttack
DiffAttack | 44.39
**31.25** | **-13.14** |
| DeiT-S | 73.63 | AdjAttack
DiffAttack | 43.18
**32.81** | **-10.37** |

of models by over 20% under $\ell_\infty$ attack with $\epsilon = 8/255$ (70.64% → 46.88% on WideResNet-28-10 and 71.29% → 45.31% on WideResNet-70-16). The effectiveness of DiffAttack also generalizes well to large-scale datasets ImageNet as shown in Table 2. Note that the robust accuracy of the state-of-the-art non-diffusion-based purification defenses [38, 21] achieve about 65% robust accuracy on CIFAR-10 with WideResNet-28-10 under $\ell_\infty = 8/255$ attack ($\epsilon = 8/255$), while the performance of score-based purification under AdjAttack in the same setting is 70.64%. However, given the strong DiffAttack, the robust accuracy of score-based purification drops to 46.88%. It motivates us to think of more effective techniques to further improve the robustness of diffusion-based purification in future work.

## 4.3 Attack against DDPM-based purification

Another line of diffusion-based purification defenses [4, 55, 56] leverages DDPM [46] to purify the images with intentionally crafted perturbations. Since backpropagating the gradients along the diffusion and sampling process with a relatively large diffusion length is unrealistic due to the large memory cost, BPDA attack [2] is adopted as the strong attack against the DDPM-based purification. However, with our proposed segment-wise forwarding-backwarding algorithm, we can compute the gradients within a small budget of memory cost, and to our best knowledge, this is the *first* work to achieve adaptive gradient-based adversarial attacks against DDPM-based purification. We compare DiffAttack with Diff-BPDA attack [4] on CIFAR-10, and the results in Table 3 demonstrate that DiffAttack outperforms the baseline by a large margin under both $\ell_\infty$ and $\ell_2$ attacks.

## 4.4 Comparison with other adaptive attack methods

Besides the AdjAttack and Diff-BPDA attacks against existing diffusion-based purification defenses, we also compare DiffAttack with other general types of adaptive attacks: 1) **black-box attack** SPSA [53] and 2) square attack [1], as well as 3) adaptive attack against score-based generative models **joint attack** (Score / Full) [60]. SPSA attack approximates the gradients by randomly sampling from a pre-defined distribution and using the finite-difference method. Square attack heuristically searches for adversarial examples in a

Table 4: Robust accuracy (%) of DiffAttack compared with other attack methods on CIFAR-10 with WideResNet-28-10 under $\ell_\infty$ attack ($\epsilon = 8/255$).

| Method | Score-based | DDPM-based |
|---|---|---|
| SPSA | 83.37 | 81.29 |
| Square Attack | 82.81 | 81.68 |
| Joint Attack (Score) | 72.74 | – |
| Joint Attack (Full) | 77.83 | 76.26 |
| Diff-BPDA | 78.13 | 75.00 |
| AdjAttack | 70.64 | – |
| **DiffAttack** | **46.88** | **54.69** |

low-dimensional space with the constraints of perturbation patterns. Joint attack (score) updates the input by the average of the classifier gradient and the output of the score estimation network, while joint attack (full) leverages the classifier gradients and the difference between the input and the purified samples. The results in Table 4 show that *DiffAttack outperforms SPSA, square attack, and joint attack by a large margin on score-based and DDPM-based purification defenses*. Note that joint attack (score) cannot be applied to the DDPM-based pipeline due to the lack of a score estimator. AdjAttack fails on the DDPM-based pipeline since it can only calculate gradients through SDE.

## 4.5 Robustness with different diffusion lengths

We observe that the diffusion length plays an extremely important role in the effectiveness of adversarial purification. Existing DDPM-based purification works [56, 55] prefer a small diffusion length, but we find it vulnerable under our DiffAttack. The influence of the diffusion length $T$ on the performance (clean/robust accuracy) of the purification defense methods is illustrated in Figure 2. We observe that *1) the clean accuracy of the purification*

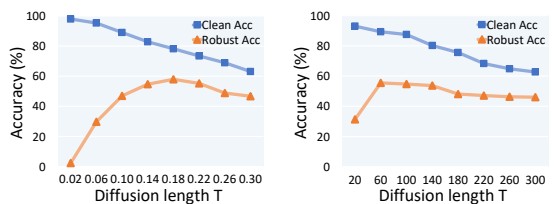

(a) Score-based purification  (b) DDPM-based purification

Figure 2: The clean/robust accuracy (%) of diffusion-based purification with different diffusion length $T$ under DiffAttack on CIFAR-10 with WideResNet-28-10 under $\ell_\infty$ attack ($\epsilon = 8/255$).

*defenses negatively correlates with the diffusion lengths* since the longer diffusion process adds more noise to the input and induces inaccurate reconstruction of the input sample; and *2) a moderate diffusion length benefits the robust accuracy* since diffusion-based purification with a small length makes it easier to compute the gradients for attacks, while models with a large diffusion length have poor clean accuracy that deteriorates the robust accuracy. We also validate the conclusion on ImageNet in Appendix D.3.

## 4.6 Comparison of memory cost

Recent work [4] computes the gradients of the diffusion and sampling process to perform the gradient-based attack, but it only considers a small diffusion length (e.g., 14 on CIFAR-10). They construct the computational graph once and for all, which is extremely expensive for memory cost with a large diffusion length. We use a segment-wise forwarding-backwarding algorithm in Section 3.3 to avoid allocating the memory for the whole computational graph. In this part, we validate the memory efficiency of our approach compared to [4]. The results

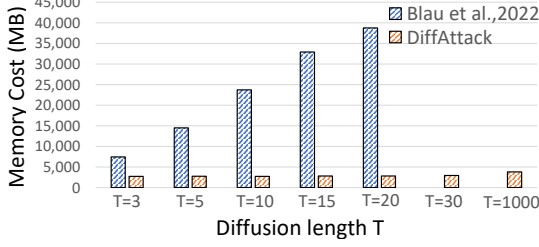

Figure 3: Comparison of memory cost of gradient backpropagation between [4] and DiffAttack with batch size 16 on CIFAR-10 with WideResNet-28-10 under $\ell_\infty$ attack.

in Figure 3 demonstrate that 1) the gradient backpropagation of [4] has the memory cost linearly correlated to the diffusion length and does not scale up to the diffusion length of 30, while 2) DiffAttack has almost constant memory cost and is able to scale up to extremely large diffusion length ($T = 1000$). The evaluation is done on an RTX A6000 GPU. In Appendix D.3, we provide comparisons of *runtime* between DiffAttack and [4] and demonstrate that DiffAttack reduces the memory cost with comparable runtime.

## 4.7 Influence of applying the deviated-reconstruction loss at different time steps

We also show that the time steps at which we apply the deviated-reconstruction loss also influence the effectiveness of DiffAttack. Intuitively, the loss added at small time steps does not suffer from vanishing/exploding gradients but lacks supervision at consequent time steps, while the loss added at large time steps gains strong supervision but suffers from the gradient problem. The results in Figure 4 show that adding deviated-reconstruction loss to uniformly sampled time steps (Uni(0,T)) achieves the best attack performance and tradeoff compared with that of adding loss to the same number of partial time steps only at the initial stage ($(0, T/3)$) or the final stage ($(2T/3, T)$). For fair comparisons, we uniformly sample $T/3$ time steps (identical to partial stage guidance $(0, T/3)$, $(2T/3, T)$) to impose $\mathcal{L}_{\text{dev}}$.

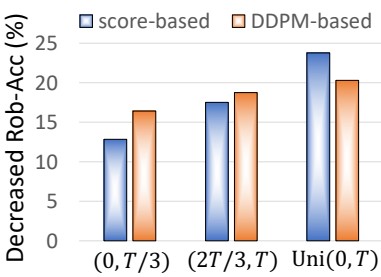

Figure 4: The impact of applying $\mathcal{L}_{dev}$ at different time steps on decreased robust accuracy (%). $T$ is the diffusion length and Uni$(0, T)$ represents uniform sampling.

## 5 Related Work

**Adversarial purification** methods purify the adversarial input before classification with generative models. Defense-gan [43] trains a GAN to restore the clean samples. Pixeldefend [48] utilizes an autoregressive model to purify adversarial examples. Another line of research [50, 22, 17, 24, 60] leverages energy-based model (EBM) and Markov chain Monte Carlo (MCMC) to perform the purification. More recently, diffusion models have seen wide success in image generation [15, 40, 41, 42, 31, 39]. They are also used to adversarial purification [34, 4, 62, 57, 51, 55, 56] and demonstrated to achieve the state-of-the-art robustness. In this work, we propose DiffAttack specifically against diffusion-based purification and show the effectiveness in different settings, which motivates future work to improve the robustness of the pipeline.

**Adversarial attacks** search for visually imperceptible signals which can significantly perturb the prediction of models [52, 20]. Different kinds of defense methods are progressively broken by advanced attack techniques, including white-box attack [6, 2, 32] and black-box attack [1, 53, 35]. [11, 12, 37, 59, 29] propose a systematic and automatic framework to attack existing defense methods. Despite attacking most defense methods, these approaches are shown to be ineffective against the diffusion-based purification pipeline due to the problem of vanishing/exploding gradient, memory cost, and randomness. Therefore, we propose DiffAttack to specifically tackle the challenges and successfully attack the diffusion-based purification defenses.

## 6 Conclusion

In this paper, we propose DiffAttack, including the deviated-reconstruction loss added on intermediate samples and a segment-wise forwarding-backwarding algorithm. We empirically demonstrate that DiffAttack outperforms existing adaptive attacks against diffusion-based purification by a large margin. We conduct a series of ablation studies and show that a moderate diffusion length benefits the model robustness, and attacks with the deviated-reconstruction loss added over uniformly sampled time steps outperform that added over only initial/final time steps, which will help to better understand the properties of diffusion process and motivate future defenses.

**Acknolwdgement.** This work is partially supported by the National Science Foundation under grant No. 1910100, No. 2046726, No. 2229876, DARPA GARD, the National Aeronautics and Space Administration (NASA) under grant no. 80NSSC20M0229, the Alfred P. Sloan Fellowship, and the Amazon research award.

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

# A Broader Impact and Limitations

**Broader impact**. As an effective and popular way to explore the vulnerabilities of ML models, adversarial attacks have been widely studied. However, recent diffusion-based purification is shown hard to attack based on different trials, which raises an interesting question of whether it can be attacked. Our paper provides the first effective attack against such defenses to identify the vulnerability of diffusion-based purification for the community and inspire more effective defense approaches. In particular, we propose an effective evasion attack against diffusion-based purification defenses which consists of a deviated-reconstruction loss at intermediate diffusion steps to induce inaccurate density gradient estimation and a segment-wise forwarding-backwarding algorithm to achieve memory-efficient gradient backpropagation. The effectiveness of the deviated-reconstruction loss helps us to better understand the properties of diffusion purification. Concretely, there exist adversarial regions in the intermediate sample space where the score approximation model outputs inaccurate density gradients and finally misleads the prediction. The observation motivates us to design a more robust sampling process in the future, and one potential way is to train a more robust score-based model. Furthermore, the segment-wise forwarding-backwarding algorithm tackles the memory issue of gradient propagation through a long path. It can be applied to the gradient calculation of any discrete Markov process almost within a constant memory cost. To conclude, our attack motivates us to rethink the robustness of a line of SOTA diffusion-based purification defenses and inspire more effective defenses.

**Limitations**. In this paper, we propose an effective attack algorithm DiffAttack against diffusion-based purification defenses. A possible negative societal impact may be the usage of DiffAttack in safety-critical scenarios such as autonomous driving and medical imaging analysis to mislead the prediction of machine learning models. However, the foundation of DiffAttack and important findings about the diffusion process properties can benefit our understanding of the vulnerabilities of diffusion-based purification defenses and therefore motivate more effective defenses in the future. Concretely, the effectiveness of DiffAttack indicates that there exist adversarial regions in the intermediate sample space where the score approximation model outputs inaccurate density gradients and finally misleads the prediction. The observation motivates us to design a more robust sampling process in the future, and one potential way is to train a more robust score-based model. Furthermore, to control a robust sampling process, it is better to provide guidance across uniformly sampled time steps rather than only at the final stage according to our findings.

---

**Algorithm 1** Segment-wise forwarding-backwarding algorithm (PyTorch-like pseudo-codes)

---

1: **Input:** $f_r, f_d, \partial \mathcal{L}/\partial \mathbf{x}'_0, \mathbf{x}_i, \mathbf{x}'_i \ (i \in [T])$
2: **Output:** $\partial \mathcal{L}/\partial \mathbf{x}_0$
3: **for** $t = 1$ **to** $T$ **do**
4:     $Creat\_Graph(f_r(\mathbf{x}'_t) \rightarrow \mathbf{x}'_{t-1})$
5:     $\mathcal{L}' \leftarrow (\partial \mathcal{L}/\partial \mathbf{x}'_{t-1}) (f_r(\mathbf{x}'_t))$
6:     $\partial \mathcal{L}/\partial \mathbf{x}'_t \leftarrow auto\_grad(\mathcal{L}', \mathbf{x}'_t)$
7:     $Release\_Graph(f_r(\mathbf{x}'_t) \rightarrow \mathbf{x}'_{t-1})$
8: **end for**
9: $\partial \mathcal{L}/\partial \mathbf{x}_T \leftarrow \partial \mathcal{L}/\partial \mathbf{x}'_T$
10: **for** $t = T - 1$ **to** $0$ **do**
11:     $Creat\_Graph(f_d(\mathbf{x}_t) \rightarrow \mathbf{x}_{t+1})$
12:     $\mathcal{L}' \leftarrow (\partial \mathcal{L}/\partial \mathbf{x}_{t+1}) (f_d(\mathbf{x}_t))$
13:     $\partial \mathcal{L}/\partial \mathbf{x}_t \leftarrow auto\_grad(\mathcal{L}', \mathbf{x}_t)$
14:     $Release\_Graph(f_d(\mathbf{x}_t) \rightarrow \mathbf{x}_{t+1})$
15: **end for**

---

# B Efficient Gradient Backpropagation

In this section, we provide the PyTorch-like pseudo-codes of the segment-wise forwarding-backwarding algorithm. At each time step $t$ in the reverse process, we only need to store the gradient $\partial \mathcal{L}/\partial \mathbf{x}'_t$, the intermediate sample $\mathbf{x}'_{t+1}$ and the model $f_r$ to construct the computational graph. When we backpropagate the gradients at the next time step $t + 1$, the computational graph at

time step $t$ will no longer be reused, and thus, we can release the memory of the graph at time step $t$. Therefore, we only have one segment of the computational graph used for gradient backpropagation in the memory at each time step. We can similarly backpropagate the gradients in the diffusion process.

## C  Proofs

### C.1  Proof of Thm. 1

**Assumption C.1.** *Consider adversarial sample $\tilde{\mathbf{x}}_0 := \mathbf{x}_0 + \delta$, where $\mathbf{x}_0$ is the clean example and $\delta$ is the perturbation. $p_t(\mathbf{x}), p'_t(\mathbf{x}), q_t(\mathbf{x}), q'_t(\mathbf{x})$ are the distribution of $\mathbf{x}_t, \mathbf{x}'_t, \tilde{\mathbf{x}}_t, \tilde{\mathbf{x}}'_t$ where $\mathbf{x}'_t$ represents the reconstruction of $\mathbf{x}_t$ at time step $t$ in the reverse process. We consider a score-based diffusion model with a well-trained score-based model $\mathbf{s}_\theta$ parameterized by $\theta$ with the clean training distribution. Therefore, we assume that $\mathbf{s}_\theta$ can achieve a low score-matching loss given a clean sample and reconstruct it in the reverse process:*

$$\|\nabla_{\mathbf{x}} \log p_t(\mathbf{x}) - \mathbf{s}_\theta(\mathbf{x}, t)\|_2^2 \leq L_u \tag{12}$$

$$D_{TV}(p_t, p'_t) \leq \epsilon_{re} \tag{13}$$

*where $D_{TV}(\cdot, \cdot)$ is the total variation distance. $L_u$ and $\epsilon_{re}$ are two small constants that characterize the score-matching loss and the reconstruction error.*

**Assumption C.2.** *We assume the score function of data distribution is bounded by $M$:*

$$\|\nabla_{\mathbf{x}} \log p_t(\mathbf{x})\|_2 \leq M, \ \|\nabla_{\mathbf{x}} \log q_t(\mathbf{x})\|_2 \leq M \tag{14}$$

**Lemma C.1.** *Consider adversarial sample $\tilde{\mathbf{x}}_0 := \mathbf{x}_0 + \delta$, where $\mathbf{x}_0$ is the clean example and $\delta$ is the perturbation. $p_t(\mathbf{x}), p'_t(\mathbf{x}), q_t(\mathbf{x}), q'_t(\mathbf{x})$ are the distribution of $\mathbf{x}_t, \mathbf{x}'_t, \tilde{\mathbf{x}}_t, \tilde{\mathbf{x}}'_t$ where $\mathbf{x}'_t$ represents the reconstruction of $\mathbf{x}_t$ in the reverse process. Given a VP-SDE parameterized by $\beta(\cdot)$ and the score-based model $\mathbf{s}_\theta$ with Assumption C.2, we have:*

$$D_{KL}(p'_t, q'_t) = \frac{1}{2} \int_t^T \beta(s) \mathbb{E}_{\mathbf{x}|\mathbf{x}_0} \|\nabla_{\mathbf{x}} \log p'_s(\mathbf{x}) - \nabla_{\mathbf{x}} \log q'_s(\mathbf{x})\|_2^2 ds + 4M^2 \int_t^T \beta(s) ds \tag{15}$$

*Proof.* The reverse process of VP-SDE can be formulated as follows:

$$d\mathbf{x} = f_{rev}(\mathbf{x}, t, p_t)dt + g_{rev}(t)d\mathbf{w}, \text{ where } f_{rev}(\mathbf{x}, t, p_t) = -\frac{1}{2}\beta(t)\mathbf{x} - \beta(t)\nabla_{\mathbf{x}} \log p_t(\mathbf{x}), \ g_{rev}(t) = \sqrt{\beta(t)} \tag{16}$$

Using the Fokker-Planck equation [44] in Equation (16), we have:

$$\frac{\partial p'_t(\mathbf{x})}{\partial t} = -\nabla_{\mathbf{x}} \left( f_{rev}(\mathbf{x}, t, p_t) p'_t(\mathbf{x}) - \frac{1}{2} g_{rev}^2(t) \nabla_{\mathbf{x}} p'_t(x) \right) \tag{17}$$

$$= \nabla_{\mathbf{x}} \left( \left( \frac{1}{2} g_{rev}^2(t) \nabla_{\mathbf{x}} \log p'_t(\mathbf{x}) - f_{rev}(\mathbf{x}, t, p_t) \right) p'_t(\mathbf{x}) \right) \tag{18}$$

Similarly, applying the Fokker-Planck equation on the reverse SDE for $q'_t(\mathbf{x})$, we can get:

$$\frac{\partial q'_t(\mathbf{x})}{\partial t} = \nabla_{\mathbf{x}} \left( \left( \frac{1}{2} g_{rev}^2(t) \nabla_{\mathbf{x}} \log q'_t(\mathbf{x}) - f_{rev}(\mathbf{x}, t, q_t) \right) q'_t(\mathbf{x}) \right) \tag{19}$$

We use the notation $h_p(\mathbf{x}) = \frac{1}{2} g_{rev}^2(t) \nabla_{\mathbf{x}} \log p'_t(\mathbf{x}) - f_{rev}(\mathbf{x}, t, p_t)$ and $h_q(x) = \frac{1}{2} g_{rev}^2(t) \nabla_{\mathbf{x}} \log q'_t(\mathbf{x}) - f_{rev}(\mathbf{x}, t, q_t)$. Then according to [34](Theorem A.1), under the assumption that $p'_t(\mathbf{x})$ and $q'_t(\mathbf{x})$ are smooth and fast decaying (i.e., $\lim_{\mathbf{x}_i \to \infty}[p'_t(\mathbf{x}) \partial \log p'(\mathbf{x})/\partial \mathbf{x}_i] = 0, \lim_{\mathbf{x}_i \to \infty}[q'_t(\mathbf{x}) \partial \log q'(\mathbf{x})/\partial \mathbf{x}_i] = 0$), we have:

$$\frac{\partial D_{KL}(p'_t, q'_t)}{\partial t} = -\int p'_t(x)[h_p(\mathbf{x}, t) - h_q(\mathbf{x}, t)]^T [\nabla_{\mathbf{x}} \log p'_t(\mathbf{x}) - \nabla_{\mathbf{x}} \log q'_t(\mathbf{x})] dx \tag{20}$$

Plugging in Equations (18) and (19), we have:

$$\frac{\partial D_{KL}(p'_t, q'_t)}{\partial t} = -\int p'_t(\mathbf{x})(\frac{1}{2}g^2_{rev}(t)\|\nabla_{\mathbf{x}}\log p'_t(\mathbf{x}) - \nabla_{\mathbf{x}}\log q'_t(\mathbf{x})\|^2_2$$
$$+ \beta(t)[\nabla_{\mathbf{x}}\log p_t(\mathbf{x}) - \nabla_{\mathbf{x}}\log q_t(\mathbf{x})]^T[\nabla_{\mathbf{x}}\log p'_t(\mathbf{x}) - \nabla_{\mathbf{x}}\log q'_t(\mathbf{x})])d\mathbf{x} \tag{21}$$

Finally, we can derive as follows:

$$D_{KL}(p'_t, q'_t) = \int_t^T \int_{\mathcal{X}} (p'_s(\mathbf{x})(\frac{1}{2}g^2_{rev}(s)\|\nabla_{\mathbf{x}}\log p'_s(\mathbf{x}) - \nabla_{\mathbf{x}}\log q'_s(\mathbf{x})\|^2_2 \tag{22}$$
$$+ \beta(s)[\nabla_{\mathbf{x}}\log p_s(\mathbf{x}) - \nabla_{\mathbf{x}}\log q_s(\mathbf{x})]^T[\nabla_{\mathbf{x}}\log p'_s(\mathbf{x}) - \nabla_{\mathbf{x}}\log q'_s(\mathbf{x})]))d\mathbf{x}ds \tag{23}$$

$$\leq \int_t^T (\frac{1}{2}g^2_{rev}(s)\mathbb{E}_{\mathbf{x}|\mathbf{x}_0}\|\nabla_{\mathbf{x}}\log p'_s(\mathbf{x}) - \nabla_{\mathbf{x}}\log q'_s(\mathbf{x})\|^2_2 + 4\beta(s)M^2)ds \tag{24}$$

$$= \frac{1}{2}\int_t^T \beta(s)\mathbb{E}_{\mathbf{x}|\mathbf{x}_0}\|\nabla_{\mathbf{x}}\log p'_s(\mathbf{x}) - \nabla_{\mathbf{x}}\log q'_s(\mathbf{x})\|^2_2 ds + 4M^2\int_t^T \beta(s)ds \tag{25}$$

$$\square$$

**Theorem 2** (Thm. 1 in the main text). *Consider adversarial sample $\tilde{\mathbf{x}}_0 := \mathbf{x}_0 + \delta$, where $\mathbf{x}_0$ is the clean example and $\delta$ is the perturbation. $p_t(\mathbf{x}), p'_t(\mathbf{x}), q_t(\mathbf{x}), q'_t(\mathbf{x})$ are the distribution of $\mathbf{x}_t, \mathbf{x}'_t, \tilde{\mathbf{x}}_t, \tilde{\mathbf{x}}'_t$ where $\mathbf{x}'_t$ represents the reconstruction of $\mathbf{x}_t$ in the reverse process. $D_{TV}(\cdot, \cdot)$ measures the total variation distance. Given a VP-SDE parameterized by $\beta(\cdot)$ and the score-based model $\mathbf{s}_\theta$ with mild assumptions that $\|\nabla_{\mathbf{x}}\log p_t(\mathbf{x}) - \mathbf{s}_\theta(\mathbf{x}, t)\|^2_2 \leq L_u$, $D_{TV}(p_t, p'_t) \leq \epsilon_{re}$, and a bounded score function by $M$ (specified with details in Appendix C.1), we have:*

$$D_{TV}(q_t, q'_t) \leq \frac{1}{2}\sqrt{\mathbb{E}_{t,\mathbf{x}|\mathbf{x}_0}\|\mathbf{s}_\theta(\mathbf{x}, t) - \nabla_{\mathbf{x}}\log q'_t(\mathbf{x})\|^2_2 + C_1}$$
$$+ \sqrt{2 - 2\exp\{-C_2\|\delta\|^2_2\}} + \epsilon_{re} \tag{26}$$

*where $C_1 = (L_u + 8M^2)\int_t^T \beta(t)dt$, $C_2 = \frac{1}{8(1 - \Pi_{s=1}^t(1 - \beta_s))}$.*

*Proof.* Since we consider VP-SDE here, we have:

$$f(\mathbf{x}, t) = -\frac{1}{2}\beta(t)\mathbf{x}, \quad g(t) = \sqrt{\beta(t)} \tag{27}$$

$$f_{rev}(\mathbf{x}, t) = -\frac{1}{2}\beta(t)\mathbf{x} - \beta(t)\nabla_{\mathbf{x}}\log p_t(\mathbf{x}), \quad g_{rev}(t) = \sqrt{\beta(t)} \tag{28}$$

Using the triangular inequality, the total variation distance between $q_t$ and $q'_t$ can be decomposed as:

$$D_{TV}(q_t, q'_t) \leq D_{TV}(q_t, p_t) + D_{TV}(p_t, p'_t) + D_{TV}(q'_t, p'_t) \tag{29}$$

According to Assumption C.1, we have $D_{TV}(p_t, p'_t) \leq \epsilon_{re}$ and thus, we only need to upper bound $D_{TV}(q_t, p_t)$ and $D_{TV}(q'_t, p'_t)$, respectively.

Using the notation $\alpha_t := 1 - \beta(t)$ and $\bar{\alpha}_t := \Pi_{s=1}^t\alpha_s$, we have:

$$\mathbf{x}_t \sim p_t := \mathcal{N}(\mathbf{x}_t; \sqrt{\bar{\alpha}_t}\mathbf{x}_0, (1 - \bar{\alpha}_t)\mathbf{I}), \quad \tilde{\mathbf{x}}_t \sim q_t := \mathcal{N}(\tilde{\mathbf{x}}_t; \sqrt{\bar{\alpha}_t}\tilde{\mathbf{x}}_0, (1 - \bar{\alpha}_t)\mathbf{I}) \tag{30}$$

Therefore, we can upper bound the total variation distance between $q_t$ and $p_t$ as follows:

$$D_{TV}(q_t, p_t) \overset{(a)}{\leq} \sqrt{2}H(\mathbf{x}_t, \tilde{\mathbf{x}}_t) \tag{31}$$

$$\overset{(b)}{=} \sqrt{2}\sqrt{1 - \exp\{-\frac{1}{8(1 - \bar{\alpha}_t)}\delta^T\delta\}} \tag{32}$$

$$= \sqrt{2 - 2\exp\{-\frac{1}{8(1 - \bar{\alpha}_t)}\|\delta\|^2_2\}} \tag{33}$$

where we leverage the inequality between the Hellinger distance $H(\cdot, \cdot)$ and total variation distance in Equation (31)(a) and we plug in the closed form of Hellinger distance between two Gaussian distribution [14] parameterized by $\mu_1, \Sigma_1, \mu_2, \Sigma_2$ in Equation (32)(b):

$$H(\mathcal{N}(\mu_1, \Sigma_1), \mathcal{N}(\mu_2, \Sigma_2))^2 = 1 - \frac{det(\Sigma_1)^{1/4} det(\Sigma_2)^{1/4}}{det\left(\frac{\Sigma_1 + \Sigma_2}{2}\right)^{1/2}} \exp\{-\frac{1}{8}(\mu_1 - \mu_2)^T \left(\frac{\Sigma_1 + \Sigma_2}{2}\right)^{-1} (\mu_1 - \mu_2)\}$$

(34)

Then, we will upper bound $D_{TV}(p_t', q_t')$. We first leverage Pinker's inequality to upper bound the total variation distance with the KL-divergence:

$$D_{TV}(p_t', q_t') \leq \sqrt{\frac{1}{2} D_{KL}(p_t', q_t')}$$

(35)

Then we plug in the results in Lemma C.1 to upper bound $KL(p_t', q_t')$ and it follows that:

$$D_{TV}(p_t', q_t') \tag{36}$$

$$\leq \sqrt{\frac{1}{2} D_{KL}(p_t', q_t')} \tag{37}$$

$$\leq \sqrt{\frac{1}{4} \int_t^T \beta(s) \mathbb{E}_{\mathbf{x}|\mathbf{x}_0} \|\nabla_{\mathbf{x}} \log p_s'(\mathbf{x}) - \nabla_{\mathbf{x}} \log q_s'(\mathbf{x})\|_2^2 ds + 2M^2 \int_t^T \beta(s) ds} \tag{38}$$

$$\leq \sqrt{\frac{1}{4} \int_t^T \beta(s) \mathbb{E}_{\mathbf{x}|\mathbf{x}_0}[\|\nabla_{\mathbf{x}} \log p_s'(\mathbf{x}) - \boldsymbol{s}_\theta(\mathbf{x}, s)\|_2^2 + \|\boldsymbol{s}_\theta(\mathbf{x}, s) - \nabla_{\mathbf{x}} \log q_s'(\mathbf{x})\|_2^2] ds + 2M^2 \int_t^T \beta(s) ds} \tag{39}$$

$$\stackrel{(a)}{\leq} \sqrt{(\frac{L_u}{4} + 2M^2) \int_t^T \beta(s) ds + \frac{1}{4} \mathbb{E}_{t, \mathbf{x}|\mathbf{x}_0} \|\boldsymbol{s}_\theta(\mathbf{x}, t) - \nabla_{\mathbf{x}} \log q_t'(\mathbf{x})\|_2^2} \tag{40}$$

where in Equation (40)(a), we leverage the fact that $\beta(\cdot)$ is bounded in $[0, 1]$.

Combining Equations (29), (33) and (40), we can finally get:

$$D_{TV}(q_t, q_t') \leq \sqrt{\frac{1}{4} \mathbb{E}_{t, \mathbf{x}|\mathbf{x}_0} \|\boldsymbol{s}_\theta(\mathbf{x}, t) - \nabla_{\mathbf{x}} \log q_t'(\mathbf{x})\|_2^2 + C_1} + \sqrt{2 - 2\exp\{-C_2 \|\delta\|_2^2\}} + \epsilon_{re}$$

(41)

where $C_1 = (\frac{L_u}{4} + 2M^2) \int_t^T \beta(s) ds$ and $C_2 = \frac{1}{8(1 - \Pi_{s=1}^t (1 - \beta_s))}$. $\qquad \square$

### C.2 Connection between the deviated-reconstruction loss and the density gradient loss for DDPM

**Theorem 3.** *Consider adversarial sample $\tilde{\mathbf{x}}_0 := \mathbf{x}_0 + \delta$, where $\mathbf{x}_0$ is the clean example and $\delta$ is the perturbation. $p_t(\mathbf{x}), p_t'(\mathbf{x}), q_t(\mathbf{x}), q_t'(\mathbf{x})$ are the distribution of $\mathbf{x}_t, \mathbf{x}_t', \tilde{\mathbf{x}}_t, \tilde{\mathbf{x}}_t'$ where $\mathbf{x}_t'$ represents the reconstruction of $\mathbf{x}_t$ in the reverse process. Given a DDPM parameterized by $\beta(\cdot)$ and the function approximator $\boldsymbol{s}_\theta$ with the mild assumptions that $\|\boldsymbol{s}_\theta(\mathbf{x}, t) - \boldsymbol{\epsilon}(\mathbf{x}_t, t)\|_2^2 \leq L_u$, $D_{TV}(p_t, p_t') \leq \epsilon_{re}$, and a bounded score function by $M$ (i.e., $\|\boldsymbol{\epsilon}(\mathbf{x}, t)\|_2 \leq M$) where $\boldsymbol{\epsilon}(\cdot, \cdot)$ represents the mapping function of the true perturbation, we have:*

$$D_{TV}(q_t, q_t') \leq \sqrt{2 - 2\exp\{-C_2 \left(\sum_{k=t+1}^T \lambda(k, t) \|\boldsymbol{s}_\theta(\tilde{\mathbf{x}}_k', k) - \boldsymbol{\epsilon}(\tilde{\mathbf{x}}_k', k)\|_2 + C_1 \|\delta\|_2 + (\sqrt{L_u} + 2M) \sum_{k=t+1}^T \lambda(k, t)\right)^2\}}$$

$$+ \sqrt{2 - 2\exp\{-\frac{1}{8}(1 - \Pi_{s=1}^t (1 - \beta_s)) \|\delta\|_2^2\}} + \epsilon_{re}$$

(42)

*where $C_1 = \left(\Pi_{s=t+1}^T \sqrt{\Pi_{k=1}^s (1 - \beta_k)}\right) \sqrt{\Pi_{s=1}^T (1 - \beta_s)}$, $C_2 = \frac{1 - \Pi_{s=1}^t (1 - \beta_s)}{8(1 - \Pi_{s=1}^{t-1}(1 - \beta_s))\beta_t}$, and*

$$\lambda(k, t) = \frac{\beta_k \Pi_{i=t+1}^{k-1} \sqrt{\Pi_{s=1}^i (1 - \beta_s)}}{\sqrt{1 - \Pi_{s=1}^k (1 - \beta_s)}}.$$

*Proof.* For ease of notation, we use the notation: $\alpha_t := 1 - \beta_t$ and $\bar{\alpha}_t := \Pi_{s=1}^t \alpha_s$. From the DDPM sampling process [25], we know that:

$$\mathbf{x}'_{t-1} \sim p'_t := \frac{1}{\sqrt{\bar{\alpha}_t}} \left( \mathbf{x}'_t - \frac{1-\alpha_t}{\sqrt{1-\bar{\alpha}_t}} \mathbf{s}_\theta(\mathbf{x}'_t, t) \right) + \sigma_t \mathbf{z} \tag{43}$$

$$\tilde{\mathbf{x}}'_{t-1} \sim q'_t := \frac{1}{\sqrt{\bar{\alpha}_t}} \left( \tilde{\mathbf{x}}'_t - \frac{1-\alpha_t}{\sqrt{1-\bar{\alpha}_t}} \mathbf{s}_\theta(\tilde{\mathbf{x}}'_t, t) \right) + \sigma_t \mathbf{z} \tag{44}$$

where $\sigma_t^2 = \frac{1-\bar{\alpha}_{t-1}}{1-\bar{\alpha}_t} \beta_t$.

$\boldsymbol{\mu}_{t,q}$ and $\boldsymbol{\mu}_{t,p}$ represent the mean of the distribution $q'_t$ and $p'_t$, respectively. Then from Equations (43) and (44), we have:

$$\boldsymbol{\mu}_{t,q} - \boldsymbol{\mu}_{t,p} = \frac{1}{\sqrt{\bar{\alpha}_t}} (\boldsymbol{\mu}_{t-1,q} - \boldsymbol{\mu}_{t-1,p}) - \frac{1-\alpha_t}{\sqrt{\bar{\alpha}_t}\sqrt{1-\bar{\alpha}_t}} (\mathbf{s}_\theta(\tilde{\mathbf{x}}'_t, t) - \mathbf{s}_\theta(\mathbf{x}'_t, t)) \tag{45}$$

Applying Equation (45) iteratively, we get:

$$\boldsymbol{\mu}_{T,q} - \boldsymbol{\mu}_{T,p} = \frac{1}{\Pi_{s=t}^T \sqrt{\bar{\alpha}_s}} (\boldsymbol{\mu}_{t-1,q} - \boldsymbol{\mu}_{t-1,p}) - \sum_{k=t}^T \frac{1-\alpha_k}{\sqrt{1-\bar{\alpha}_k}\Pi_{i=k}^T \sqrt{\bar{\alpha}_i}} (\mathbf{s}_\theta(\tilde{\mathbf{x}}'_k, k) - \mathbf{s}_\theta(\mathbf{x}'_k, k)) \tag{46}$$

On the other hand, $\boldsymbol{\mu}_{T,q} - \boldsymbol{\mu}_{T,p}$ can be formulated explicitly considering the Gaussian distribution at time step $T$ in the diffusion process:

$$\boldsymbol{\mu}_{T,q} - \boldsymbol{\mu}_{T,p} = \sqrt{\bar{\alpha}_T}(\tilde{\mathbf{x}}_0 - \mathbf{x}_0) = \sqrt{\bar{\alpha}_T}\delta \tag{47}$$

Combining Equations (46) and (47), we can derive that:

$$\|\boldsymbol{\mu}_{T,q} - \boldsymbol{\mu}_{T,p}\|_2 \tag{48}$$

$$= \left(\Pi_{s=t+1}^T \sqrt{\bar{\alpha}_s}\right) \sqrt{\bar{\alpha}_T}\|\delta\|_2 + \sum_{k=t+1}^T \frac{(1-\alpha_k)\Pi_{i=t+1}^{k-1}\sqrt{\bar{\alpha}_i}}{\sqrt{1-\bar{\alpha}_k}} \|\mathbf{s}_\theta(\tilde{\mathbf{x}}'_k, k) - \mathbf{s}_\theta(\mathbf{x}'_k, k)\|_2 \tag{49}$$

$$\leq \left(\Pi_{s=t+1}^T \sqrt{\bar{\alpha}_s}\right) \sqrt{\bar{\alpha}_T}\|\delta\|_2 + \sum_{k=t+1}^T \frac{(1-\alpha_k)\Pi_{i=t+1}^{k-1}\sqrt{\bar{\alpha}_i}}{\sqrt{1-\bar{\alpha}_k}} \left(\|\mathbf{s}_\theta(\tilde{\mathbf{x}}'_k, k) - \epsilon(\mathbf{x}'_k, k)\|_2 + \|\epsilon(\mathbf{x}'_k, k) - \mathbf{s}_\theta(\mathbf{x}'_k, k))\|_2\right) \tag{50}$$

$$\leq \left(\Pi_{s=t+1}^T \sqrt{\bar{\alpha}_s}\right) \sqrt{\bar{\alpha}_T}\|\delta\|_2 + \sqrt{L_u}\sum_{k=t+1}^T \lambda(k,t) + \sum_{k=t+1}^T \lambda(k,t)\left(\|\mathbf{s}_\theta(\tilde{\mathbf{x}}'_k, k) - \epsilon(\tilde{\mathbf{x}}'_k, k)\|_2 + \|\epsilon(\tilde{\mathbf{x}}'_k, k) - \epsilon(\mathbf{x}'_k, k)\|_2\right) \tag{51}$$

$$\leq \left(\Pi_{s=t+1}^T \sqrt{\bar{\alpha}_s}\right) \sqrt{\bar{\alpha}_T}\|\delta\|_2 + (\sqrt{L_u} + 2M)\sum_{k=t+1}^T \lambda(k,t) + \sum_{k=t+1}^T \lambda(k,t)\|\mathbf{s}_\theta(\tilde{\mathbf{x}}'_k, k) - \epsilon(\tilde{\mathbf{x}}'_k, k)\|_2 \tag{52}$$

where $\lambda(k,t) = \frac{(1-\alpha_k)\Pi_{i=t+1}^{k-1}\sqrt{\bar{\alpha}_i}}{\sqrt{1-\bar{\alpha}_k}}$.

We then leverage the closed form formulation of the Hellinger distance between two Gaussian distributions [14] parameterized by $\mu_1, \Sigma_1, \mu_2, \Sigma_2$:

$$H^2(\mathcal{N}(\mu_1, \Sigma_1), \mathcal{N}(\mu_2, \Sigma_2)) = 1 - \frac{det(\Sigma_1)^{1/4}det(\Sigma_2)^{1/4}}{det\left(\frac{\Sigma_1 + \Sigma_2}{2}\right)^{1/2}} \exp\{-\frac{1}{8}(\mu_1 - \mu_2)^T \left(\frac{\Sigma_1 + \Sigma_2}{2}\right)^{-1}(\mu_1 - \mu_2)\} \tag{53}$$

Applying it to distribution $p'_t$ and $q'_t$, we have:

$$H^2(p'_t, q'_t) = 1 - \exp\{-\frac{1-\bar{\alpha}_t}{8(1-\bar{\alpha}_{t-1})\beta_t} \|\boldsymbol{\mu}_{t,q} - \boldsymbol{\mu}_{t,p}\|_2^2\} \tag{54}$$

$$\leq 1 - \exp\{-C_2 \left(C_1\|\delta\|_2 + (\sqrt{L_u} + 2M)\sum_{k=t+1}^T \lambda(k,t) + \sum_{k=t+1}^T \lambda(k,t)\|\mathbf{s}_\theta(\tilde{\mathbf{x}}'_k, k) - \epsilon(\tilde{\mathbf{x}}'_k, k)\|_2\right)^2\} \tag{55}$$

where $C_1 = \left(\Pi_{s=t+1}^T \sqrt{\bar{\alpha}_s}\right) \sqrt{\bar{\alpha}_T}$ and $C_2 = \dfrac{1 - \bar{\alpha}_t}{8(1 - \bar{\alpha}_{t-1})\beta_t}$. Finally, it follows that:

$$D_{TV}(q_t, q_t') \leq D_{TV}(q_t, p_t) + D_{TV}(p_t, p_t') + D_{TV}(q_t', p_t') \tag{56}$$

$$\leq \sqrt{2 - 2\exp\{-\frac{1}{8}(1 - \bar{\alpha}_t)\|\delta\|_2^2\} + \epsilon_{re}} + \sqrt{2}H(q_t', p_t') \tag{57}$$

$$\leq \sqrt{2 - 2\exp\{-C_2\left(C_1\|\delta\|_2 + (\sqrt{L_u} + 2M)\sum_{k=t+1}^T \lambda(k, t) + \sum_{k=t+1}^T \lambda(k, t)\|s_\theta(\tilde{\mathbf{x}}_k', k) - \epsilon(\tilde{\mathbf{x}}_k', k)\|_2\right)^2\}} \tag{58}$$

$$+ \sqrt{2 - 2\exp\{-\frac{1}{8}(1 - \bar{\alpha}_t)\|\delta\|_2^2\} + \epsilon_{re}} \tag{59}$$

$$= \sqrt{2 - 2\exp\{-C_2\left(\sum_{k=t+1}^T \lambda(k, t)\|s_\theta(\tilde{\mathbf{x}}_k', k) - \epsilon(\tilde{\mathbf{x}}_k', k)\|_2 + C_1\|\delta\|_2 + (\sqrt{L_u} + 2M)\sum_{k=t+1}^T \lambda(k, t)\right)^2\}} \tag{60}$$

$$+ \sqrt{2 - 2\exp\{-\frac{1}{8}(1 - \Pi_{s=1}^t(1 - \beta_s))\|\delta\|_2^2\} + \epsilon_{re}} \tag{61}$$

where $C_1 = \left(\Pi_{s=t+1}^T \sqrt{\Pi_{k=1}^s(1 - \beta_k)}\right) \sqrt{\Pi_{s=1}^T(1 - \beta_s)}$, $C_2 = \dfrac{1 - \Pi_{s=1}^t(1 - \beta_s)}{8(1 - \Pi_{s=1}^{t-1}(1 - \beta_s))\beta_t}$, and $\lambda(k, t) = \dfrac{\beta_k \Pi_{i=t+1}^{k-1}\sqrt{\Pi_{s=1}^i(1 - \beta_s)}}{\sqrt{1 - \Pi_{s=1}^k(1 - \beta_s)}}$.

$\square$

## D  Experiment

### D.1  Pseudo-code of DiffAttack

Given an input pair $(\mathbf{x}, y)$ and the perturbation budget, we notate $\mathcal{L} := \mathcal{L}_{cls} + \lambda\mathcal{L}_{dev}$ (Equation (8)) the surrogate loss, $\Pi$ the projection operator given the perturbation budget and distance metric, $\eta$ the step size, $\alpha$ the momentum coefficient, $N_{\text{iter}}$ the number of iterations, and $W$ the set of checkpoint iterations. Concretely, we select $\mathcal{L}_{cls}$ as the cross-entropy loss in the first round and DLR loss in the second round following [34]. The gradient of the surrogate loss with respect to the samples is computed by forwarding the samples and backwarding the gradients for multiple times and taking the average to tackle the problem of randomness. We also optimize all the samples, including the misclassified ones, to push them away from the decision boundary. The gradient can be computed with our segment-wise forwarding-backwarding algorithm in Section 3.3, which enables DiffAttack to be the *first* fully adaptive attack against the DDPM-based purification defense. The complete procedure is provided in Algorithm 2.

### D.2  Experiment details

We use pretrained score-based diffusion models [49] on CIFAR-10, guided diffusion models [15] on ImageNet, and DDPM [25] on CIFAR-10 to purify the images following the literature [34, 4, 55, 56]. Due to the high computational cost, we follow [34] to randomly select a fixed subset of 512 images sampled from the test set to evaluate the robust accuracy for fair comparisons. We implement DiffAttack in the framework of AutoAttack [12], and we use the same hyperparameters. Specifically, the number of iterations of attacks ($N_{\text{iter}}$) is 100, and the number of iterations to approximate the gradients (EOT) is 20. The momentum coefficient $\alpha$ is 0.75, and the step size $\eta$ is initialized with $2\epsilon$ where $\epsilon$ is the maximum $\ell_p$-norm of the perturbations. The balance factor $\lambda$ between the classification-guided loss and the deviated-reconstruction loss in Equation (8) is fixed as 1.0 and $\alpha(\cdot)$ is set the reciprocal of the size of sampled time steps in the evaluation. We consider $\epsilon = 8/255$ and $\epsilon = 4/255$ for $\ell_\infty$ attack and $\epsilon = 0.5$ for $\ell_2$ attack following the literature [11, 12].

We use randomly selected 3 seeds and report the averaged results for evaluations. CIFAR-10 is under the MIT license and ImageNet is under the BSD 3-clause license.

**More details of baselines.** In this part, we illustrate more details of the baselines 1) SOTA attacks against score-based diffusion *adjoint attack* (AdjAttack) [34], 2) SOTA attack against DDPM-based

---

**Algorithm 2** DiffAttack

---
1: **Input:** $\mathcal{L} := \mathcal{L}_{cls} + \lambda\mathcal{L}_{dev}$, $\Pi$, $(\mathbf{x}, y)$, $\eta$, $\alpha$, $N_{\text{iter}}$, $W = \{w_0, \ldots, w_n\}$
2: **Output:** $\tilde{\mathbf{x}}$
3: $\tilde{\mathbf{x}}^{(0)} \leftarrow \tilde{\mathbf{x}}$
4: $\tilde{\mathbf{x}}^{(1)} \leftarrow \Pi\left(\tilde{\mathbf{x}}^{(0)} + \eta\nabla_{\tilde{\mathbf{x}}^{(0)}}\mathcal{L}(\tilde{\mathbf{x}}^{(0)}, y)\right)$
5: $L_{\max} \leftarrow \max\{\mathcal{L}(\tilde{\mathbf{x}}^{(0)}, y), \mathcal{L}(\tilde{\mathbf{x}}^{(1)}, y)\}$
6: $\tilde{\mathbf{x}} \leftarrow \tilde{\mathbf{x}}^{(0)}$ **if** $L_{\max} \equiv \mathcal{L}(\tilde{\mathbf{x}}^{(0)}, y)$ **else** $\tilde{\mathbf{x}} \leftarrow \tilde{\mathbf{x}}^{(1)}$
7: **for** $k = 1$ **to** $N_{\text{iter}}-1$ **do**
8:     $\mathbf{z}^{(k+1)} \leftarrow \Pi\left(\tilde{\mathbf{x}}^{(k)} + \eta\nabla_{\tilde{\mathbf{x}}^{(k)}}\mathcal{L}(\tilde{\mathbf{x}}^{(k)}, y)\right)$
9:     $\tilde{\mathbf{x}}^{(k+1)} \leftarrow \Pi\left(\tilde{\mathbf{x}}^{(k)} + \alpha(\mathbf{z}^{(k+1)} - \tilde{\mathbf{x}}^{(k)}) + (1-\alpha)(\tilde{\mathbf{x}}^{(k)} - \tilde{\mathbf{x}}^{(k-1)})\right)$
10:    **if** $\mathcal{L}(\tilde{\mathbf{x}}^{(k+1)}, y) > L_{\max}$ **then**
11:        $\tilde{\mathbf{x}} \leftarrow \tilde{\mathbf{x}}^{(k+1)}$ and $L_{\max} \leftarrow \mathcal{L}(\tilde{\mathbf{x}}^{(k+1)}, y)$
12:    **end if**
13:    **if** $k \in W$ **then**
14:        $\eta \leftarrow \eta/2$
15:    **end if**
16: **end for**

---

diffusion *Diff-BPDA attack* [4], 3) SOTA black-box attack *SPSA* [53] and *square attack* [1], and 4) specific attack against EBM-based purification *joint attack* (score/full) [60]. AdjAttack selects the surrogate loss $\mathcal{L}$ as the cross-entropy loss and DLR loss and leverages the adjoint method [28] to efficiently backpropagate the gradients through SDE. The basic idea is to obtain the gradients via solving an augmented SDE. For the SDE in Equation (4), the augmented SDE that computes the gradients $\partial\mathcal{L}/\partial\mathbf{x}'_T$ of back=propagating through it is given by:

$$\begin{pmatrix} \mathbf{x}'_T \\ \frac{\partial\mathcal{L}}{\partial\mathbf{x}'_T} \end{pmatrix} = \texttt{sdeint}\left(\begin{pmatrix} \mathbf{x}'_0 \\ \frac{\partial\mathcal{L}}{\partial\mathbf{x}'_0} \end{pmatrix}, \tilde{\mathbf{f}}, \tilde{\mathbf{g}}, \tilde{\mathbf{w}}, 0, T\right) \tag{62}$$

where $\frac{\partial\mathcal{L}}{\partial\mathbf{x}'_0}$ is the gradient of the objective $\mathcal{L}$ w.r.t. the $\mathbf{x}'_0$, and

$$\begin{aligned} \tilde{\mathbf{f}}([\mathbf{x}; \mathbf{z}], t) &= \begin{pmatrix} \mathbf{f}_{\text{rev}}(\mathbf{x}, t) \\ \frac{\partial\mathbf{f}_{\text{rev}}(\mathbf{x},t)}{\partial\mathbf{x}}\mathbf{z} \end{pmatrix} \\ \tilde{\mathbf{g}}(t) &= \begin{pmatrix} -\mathbf{g}_{\text{rev}}(t)\mathbf{1}_d \\ \mathbf{0}_d \end{pmatrix} \\ \tilde{\mathbf{w}}(t) &= \begin{pmatrix} -\mathbf{w}(1-t) \\ -\mathbf{w}(1-t) \end{pmatrix} \end{aligned} \tag{63}$$

with $\mathbf{1}_d$ and $\mathbf{0}_d$ representing the $d$-dimensional vectors of all ones and all zeros, respectively and $\mathbf{f}_{\text{rev}}(\mathbf{x}, t) := -\frac{1}{2}\beta(t)\mathbf{x} - \beta(t)\nabla_{\mathbf{x}}\log p_t(\mathbf{x})$, $\mathbf{g}_{\text{rev}}(t) := \sqrt{\beta(t)}$.

SPSA attack approximates the gradients by randomly sampling from a pre-defined distribution and using the finite-difference method. Square attack heuristically searches for adversarial examples in a low-dimensional space with the constraints of the perturbation pattern (i.e., constraining the square shape of the perturbation). Joint attack (score) updates the input by the weighted average of the classifier gradient and the output of the score estimation network (i.e., the gradient of log-likelihood with respect to the input), while joint attack (full) leverages the classifier gradients and the difference between the input and the purified samples. The update of the joint attack (score) is formulated as follows:

$$\tilde{\mathbf{x}} \leftarrow \tilde{\mathbf{x}} + \eta\left(\lambda'\texttt{sign}(s_\theta(\tilde{\mathbf{x}})) + (1-\lambda')\texttt{sign}(\nabla_{\tilde{\mathbf{x}}}\mathcal{L}(F(P(\tilde{\mathbf{x}})), y))\right) \tag{64}$$

The update of the joint attack (full) is formulated as follows:

$$\tilde{\mathbf{x}} \leftarrow \tilde{\mathbf{x}} + \eta\left(\lambda'\texttt{sign}(F(P(\tilde{\mathbf{x}})) - \tilde{\mathbf{x}}) + (1-\lambda')\texttt{sign}(\nabla_{\tilde{\mathbf{x}}}\mathcal{L}(F(P(\tilde{\mathbf{x}})), y))\right) \tag{65}$$

where $\eta$ is the step size and $\lambda'$ the balance factor fixed as $0.5$ in the evaluation.

Table 5: Comparisons of gradient backpropagation time per batch(second)/Memory cost (MB) between [4] and DiffAttack. We evaluate on CIFAR-10 with WideResNet-28-10 with batch size 16.

| Method | $T=5$ | $T=10$ | $T=15$ | $T=20$ | $T=30$ | $T=1000$ |
|---|---|---|---|---|---|---|
| [4] | 0.45/14,491 | 0.83/23,735 | 1.25/32,905 | 1.80/38,771 | — | — |
| DiffAttack | 0.44/2,773 | 0.85/2,731 | 1.26/2,805 | 1.82/2,819 | 2.67/2,884 | 85.81/3,941 |

Table 6: The clean / robust accuracy (%) of diffusion-based purification with different diffusion lengths $T$ under DiffAttack. The evaluation is done on ImageNet with ResNet-50 under $\ell_\infty$ attack ($\epsilon = 4/255$).

| $T=50$ | $T=100$ | $T=150$ | $T=200$ |
|---|---|---|---|
| 71.88 / 12.46 | 68.75 / 24.62 | 67.79 / 28.13 | 65.62 / 26.83 |

## D.3 Additional Experiment Results

**Efficiency evaluation**. We evaluate the **wall clock time** per gradient backpropagation of the segment-wise forwarding-backwarding algorithm for different diffusion lengths and compare the time efficiency as well as the memory costs with the standard gradient backpropagation in previous attacks [4]. The results in Table 5 indicate that the segment-wise forwarding-backwarding algorithm consumes comparable wall clock time per gradient backpropagation compared with [4] and achieves a much better tradeoff between time efficiency and memory efficiency. The evaluation is done on an RTX A6000 GPU with 49,140 MB memory. In the segment-wise forwarding-backwarding algorithm, we require one forward pass and one backpropagation pass in total for the gradient computation, while the standard gradient backpropagation in [4] requires one backpropagation pass. However, since the backpropagation pass is much more expensive than the forward pass [36], our segment-wise forwarding-backwarding algorithm can achieve comparable time efficiency while significantly reducing memory costs.

**More ablation studies on ImageNet**. We conduct more evaluations on ImageNet to consolidate the findings in CIFAR-10. We evaluate the clean/robust accuracy (%) of diffusion-based purification with different diffusion lengths $T$ under DiffAttack. The results in Table 6 indicate that 1) the clean accuracy of the purification defenses negatively correlates with the diffusion lengths, and 2) a moderate diffusion length benefits the robust accuracy under DiffAttack.

**Tansferability of DiffAttack**. ACA [10] and Diff-PGD attack [58] explore the transferability of unrestricted adversarial attack, which generates realistic adversarial examples to fool the classifier and maintain the photorealism. They demonstrate that this kind of semantic attack transfers well to other models. To explore the transferability of adversarial examples by $\ell_p$-norm-based DiffAttack, we evaluate the adversarial examples generated on score-based purification withResNet-50 on defenses with pretrained WRN-50-2 and DeiT-S. The results in Table 7 indicate that DiffAttack also transfers better than AdjAttack and achieves much lower robust accuracy on other models.

**Ablation study of balance factor $\lambda$**. As shown in Equation (11), $\lambda$ controls the balance of the two objectives. A small $\lambda$ can weaken the deviated-reconstruction object and make the attack suffer more from the vanishing/exploded gradient problem, while a large $\lambda$ can downplay the guidance of the classification loss and confuse the direction towards the decision boundary of the classifier. The results in Table 8 show that selecting $\lambda$ as 1.0 achieves better tradeoffs empirically, so we fix it as 1.0 for experiments.

## D.4 Visualization

In this section, we provide the visualization of adversarial examples generated by DiffAttack. Based on the visualization on CIFAR-10 and ImageNet with different network architectures, we conclude that the perturbation generated by DiffAttack is stealthy and imperceptible to human eyes and hard to be utilized by defenses.

Table 7: Robust accuracy (%) with $\ell_\infty$ attack ($\epsilon = 8/255$) against score-based diffusion purification on CIFAR-10. The adversarial examples are optimized on the diffusion purification defense with pretrained ResNet-50 and evaluated on defenses with other types of models including WRN-50-2 and DeiT-S.

|            | *ResNet-50* | WRN-50-2 | DeiT-S |
|------------|-------------|----------|--------|
| AdjAttack  | 40.93       | 52.37    | 54.53  |
| DiffAttack | **28.13**   | **37.28**| **39.62** |

Table 8: The impact of different loss weights $\lambda$ on the robust accuracy (%). We perform $\ell_\infty$ ($\epsilon = 8/255$) against score-based diffusion purification on CIFAR-10 with WideResNet-28-10 and diffusion length $T = 0.1$.

| $\lambda = 0.1$ | $\lambda = 1.0$ | $\lambda = 10.0$ |
|-----------------|-----------------|------------------|
| 54.69           | **46.88**       | 53.12            |

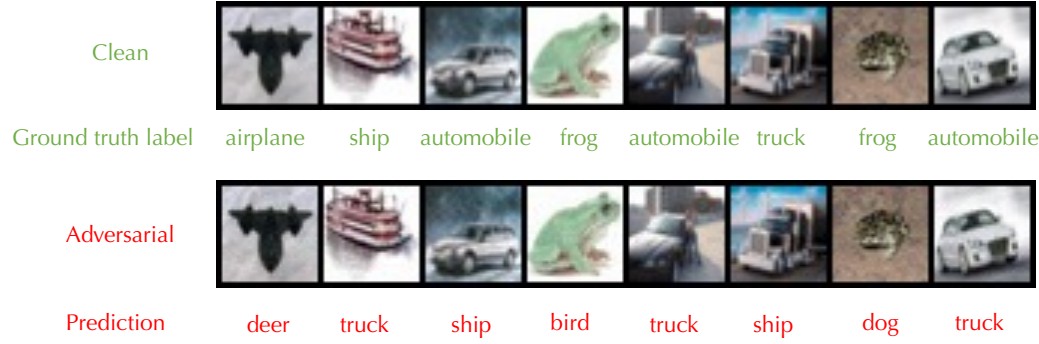

Figure 5: Visualization of the clean images and adversarial samples generated by DiffAttack on CIFAR-10 with $\ell_\infty$ attack ($\epsilon = 8/255$) against score-based purification with WideResNet-28-10.

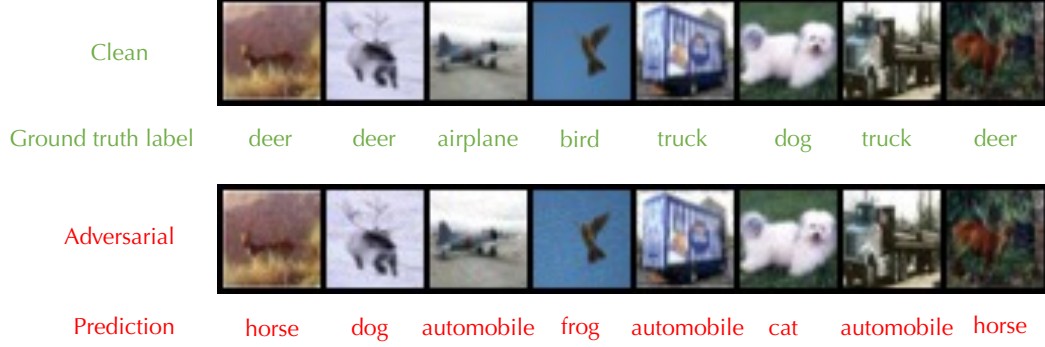

Figure 6: Visualization of the clean images and adversarial samples generated by DiffAttack on CIFAR-10 with $\ell_\infty$ attack ($\epsilon = 8/255$) against score-based purification with WideResNet-70-16.

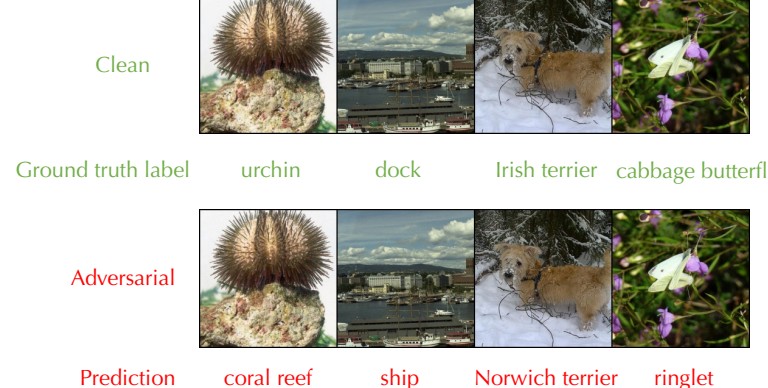

Figure 7: Visualization of the clean images and adversarial samples generated by DiffAttack on ImageNet with $\ell_\infty$ attack ($\epsilon = 4/255$) against score-based purification with WideResNet-50-2.

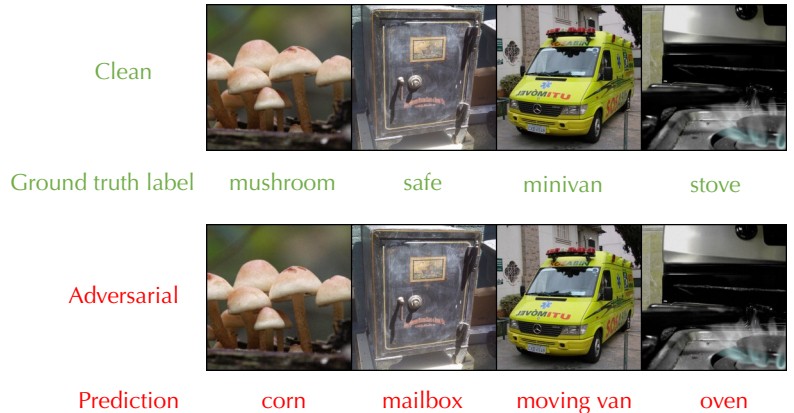

Figure 8: Visualization of the clean images and adversarial samples generated by DiffAttack on ImageNet with $\ell_\infty$ attack ($\epsilon = 4/255$) against score-based purification with DeiT-S.

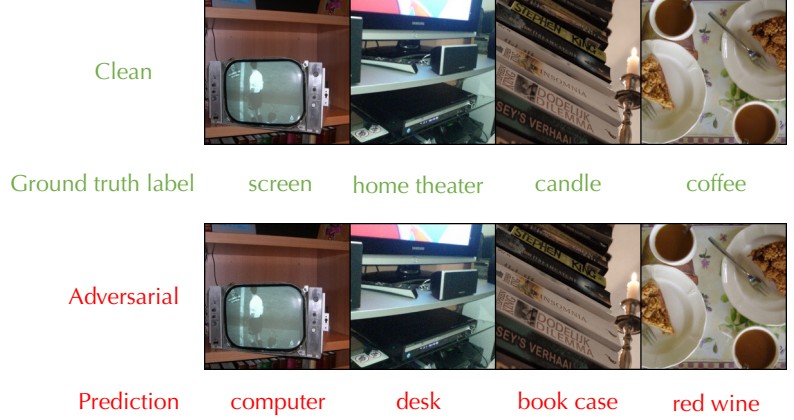

Figure 9: Visualization of the clean images and adversarial samples generated by DiffAttack on ImageNet with a larger perturbation radius: $\ell_\infty$ attack ($\epsilon = 8/255$) against score-based purification with ResNet-50.

