# OpenReview forum: "DiffAttack: Evasion Attacks Against Diffusion-Based Adversarial Purification"
_NeurIPS.cc/2023/Conference — NeurIPS 2023 poster_

### Official Review · Reviewer_Vu7G · 2023-06-28

**Soundness:** 3 good
**Presentation:** 3 good
**Contribution:** 3 good
**Rating:** 5
**Confidence:** 2

**Summary:**

This paper aims to generate adv-samples that can effectively attack diffusion-based purification. Two key strategies are proposed: (a) deviated reconstruction loss which is used as a surrogate loss to relieve gradient obfuscation existing in the long computational graph; (b) segment-wise forward-backward strategy is applied to relieve the high memory cost to calculate accurate $L_{cls}$. The experiments show that the method proposed is effective.

**Strengths:**

Overall, the paper is well-written and easy to read, the clarification is clear and

- The problem is important since diffusion-based purification is proven to be effective and model-agnostic.
- The proposed deviated reconstruction is proven to be effective.
- The segment-wise forward-backward is effective and well-design
- The experiments results are extensive.



**Weaknesses:**

I mainly have some questions here:

- Will the segment-wise forward-backward affect the speed compared with the original forward-backward?
- Intuitively, the adv-samples generated by DiffAttack are (1) hard to be purified by one specific DM, (2) hard to denoise at each stage in a given DM. When the budget $\epsilon$ is larger, e.g. 8, 16 for imagenet, what will the adv-samples be like?
- The code is not provided in the supp materials, it will be better to show some demo code
- Do well need to carefully design the $\lambda$ weights for the two losses?
- (Some related paper) Can DiffAttack also transfer better to other networks? This topic is quite related to some recent papers, which prove that only attacking the diffusion model can be transferred well. [1, 2]

I am willing to raise my score if most of the questions are solved.



[1] Chen, Z., Li, B., Wu, S., Jiang, K., Ding, S., & Zhang, W. (2023). Content-based Unrestricted Adversarial Attack. arXiv preprint arXiv:2305.10665.

[2] Xue, H., Araujo, A., Hu, B. and Chen, Y., (2023). Diffusion-Based Adversarial Sample Generation for Improved Stealthiness and Controllability. arXiv preprint arXiv:2305.16494.


**Questions:**

refer to `Weaknesses`

**Limitations:**

NULL

---

> ### Author Rebuttal · Authors · 2023-08-09
>
> > Q1: The speed of segment-wise forward-backward compared with the original forward-backward.
>
> Due to the page limits, in line 361-362, we mention that we provide comparisons of runtime between DiffAttack and [a] (standard forward-backward) in Appendix D.3. The results demonstrate that DiffAttack reduces the memory cost with comparable runtime. The reason is that we require one forward-backward pass for each segment while standard methods only require one backward pass, but since the backpropagation pass is much more expensive than the forward pass. Therefore, our segmentwise forwarding-backwarding manner demonstrates similar runtime as the original forward-backward.
>
> *[a] Blau, Tsachi, et al. "Threat model-agnostic adversarial defense using diffusion models." arXiv preprint arXiv:2207.08089 (2022).*
>
> > Q2: Visualizations of adversarial samples with a larger perturbation on ImageNet.
>
> We follow your suggestion to add a visualization of the adversarial examples by DiffAttack with a larger perturbation. The results in Figure 10 in the one-page rebuttal PDF indicate that the perturbation generated by DiffAttack is stealthy and imperceptible to human eyes and hard to be utilized by defenses.
>
> > Q3: Codes of DiffAttack.
>
> We provided the anonymous link of the code repo in line 693 in the supplementary material. We will make it more clear in the main text.
>
> > Q4: Ablation studies of weights for the two losses $\lambda$.
>
> We follow the suggestion to add ablation studies of the weights $\lambda$ to show the interplay of $\mathcal{L} _{dev}$ and $\mathcal{L} _{cls}$. According to Equation (11), $\lambda$ controls the balance of the classification-guided loss $\mathcal{L} _{cls}$ and the deviated-reconstruction loss $\mathcal{L} _{dev}$. A small $\lambda$ can weaken the deviated-reconstruction object and makes the attack suffer more from the gradient obfuscation problem, while a large $\lambda$ can downplay the guidance of the classification loss and confuse the direction towards the decision boundary of the classifier. We evaluate the effectiveness of different loss weights $\lambda$ in Table 8 in the one-page rebuttal PDF. The results show that selecting $\lambda$ as $1.0$ achieves better tradeoffs empirically. For consistency, we fix $\lambda$ as $1.0$ for all the experiments.
>
> > Q5: Transferability of DiffAttack to other networks.
>
> ACA [b] and Diff-PGD attack [c] explores the transferability of unrestricted adversarial attack, which generates realistic adversarial examples to fool the classifier and maintain the photorealism. They demonstrate that this kind of semantic attack transfers well to other models. To explore the transferability of adversarial examples by $\ell_p$-norm-based DiffAttack, we evaluate the adversarial examples generated on score-based purification withResNet-50 on defenses with pretrained WRN-50-2 and DeiT-S. The results in Table 9 in the one-page rebuttal PDF indicate that DiffAttack also transfers better than AdjAttack and achieves much lower robust accuracy on other models.
>
> *[b] Chen, Z., Li, B., Wu, S., Jiang, K., Ding, S., & Zhang, W. (2023). Content-based Unrestricted Adversarial Attack. arXiv preprint arXiv:2305.10665.*
>
> *[c] Xue, H., Araujo, A., Hu, B. and Chen, Y., (2023). Diffusion-Based Adversarial Sample Generation for Improved Stealthiness and Controllability. arXiv preprint arXiv:2305.16494.*

---

> > ### Comment · Reviewer_Vu7G · 2023-08-13
> > **Thanks for the reply**
> >
> > I have read the rebuttal by the authors, and I have also read the comments of other reviewers. I would like to remain my score and suggest a Borderline acceptance.

---

### Official Review · Reviewer_yE5n · 2023-06-30

**Soundness:** 2 fair
**Presentation:** 1 poor
**Contribution:** 2 fair
**Rating:** 6
**Confidence:** 4

**Summary:**

This paper proposes a new adaptive attack to unify the robustness evaluation of diffusion-based purification defenses, including both score-based and DDPM-based diffusion models. The new attack mainly includes two novel techniques to overcome the challenges of evaluating diffusion-based purification defenses. The first technique is a regularizer that encourages a large distance between the forward images and their counterparts in the reverse process, which is proposed to tackle the vanishing or exploding gradients problem. The second technique is a forward-backward algorithm that allows for gradient back-propagation segment by segment, which is proposed to reduce the high memory cost. Evaluations on CIFAR-10 and ImageNet show that the proposed adaptive attack can lower the robust accuracy by a large margin while maintaining low memory costs.


**Strengths:**

### Originality

* **Novel attack techniques.** The overall attack is novel. The proposed regularizer is not surprising but requires some insight to apply properly. The segment-wise forward-backward algorithm is also novel to my knowledge.

### Quality

* **Comprehensive evaluations.** The evaluation includes two models on CIFAR-10 and three models on ImageNet, with two purification defenses and several baseline adaptive attacks.
* **Good ablation studies.** Other studies of the diffusion lengths, memory cost, and the proposed loss term are relatively sufficient.
* **The theorem unifies attacks against score-based and DDPM-based purification defenses.** The presented two theorems show that increasing the new loss term can increase the data estimation loss of both DDPM-based and score-based purification defenses, although these theorems are not sufficiently discussed or motivated.

### Clarity

N/A

### Significance

* **Timely topic.** Diffusion-based purification is a fast-developing direction in adversarial example defenses. Given the well-known challenges and the lacked systemized evaluation of these defenses, this paper tackles an important problem in adversarial example defenses.
* **Strong results.** It is good to see that the proposed attack can lower the robust accuracy by around 10% to 20%. The memory cost is also very low compared with previous approaches.


**Weaknesses:**

### Originality

**1. [Moderate] Missing related work for memory-efficient gradient back-propagation.**

One major contribution of this paper is the memory-efficient gradient back-propagation algorithm. However, this paper did not discuss this literature and potential previous approaches for improving memory efficiency, either for the general optimization problem or for the diffusion model specifically. It is also claimed that this algorithm *"can be used for any discrete Markov process."* This is a strong claim and seems to contribute to a broader area. Yet, without a solid discussion of related work, it is unclear how to position this algorithm in the literature. For example, is there any existing memory-efficient algorithm that is also able to tackle the memory overheads when evaluating diffusion-based purification defense? If so, a further ablation study may be necessary to show why the algorithm in this paper is still needed. If not, it is recommended to highlight this contribution more.

**2. [Minor] Attack framework vs. attack techniques.**

The proposed methods are more like novel attack techniques rather than an attack framework. As far as I can tell from Algorithm 2, the attack generally follows the existing PGD or APGD attack framework (gradient update + projection). It is suggested to tune down this claim if the authors could not differentiate the proposed attack's framework from existing ones.

### Quality

**3. [Major] Did not address the motivated challenge of unbounded randomness.**

The authors called out the *"unbounded randomness"* problem in many places throughout the paper (e.g., lines 6, 32, 43, 133, 226, 391). However, the paper did not explain how this problem was resolved by the two proposed techniques. The missing explanation is confirmed by the summarized technical contributions in L69-83, where the two proposed techniques both tackle one of the remaining two challenges. There are two places discussing the randomness problem but likely for some other contexts: the expectation in Equation (8) and the optimization decision in L234-240. However, neither of these discussions could explain the "unbounded randomness" challenge in the original evaluation.

Please clarify if the proposed attack could tackle the randomness problem. If not, it is suggested to discuss why the proposed attack does not need to tackle the randomness problem to obtain strong results. If that is indeed the case, I am curious if the results in this paper would connect to the recent studies showing that randomness might be unhelpful in defenses [A, B]. Otherwise, is it possible to lower the robust accuracy even more if the randomness problem is explicitly tackled? Combined together they could strengthen this paper by showing some insights into the most challenging problem in evaluating purification defenses (gradient, memory, or randomn[[2023-06-28]]ess).

[A] On the Limitations of Stochastic Pre-processing Defenses. NeurIPS 2022.
[B] Randomness in ML Defenses Helps Persistent Attackers and Hinders Evaluators. arXiv 2023.

**4. [Major] It is unclear how the gradient issue was resolved.**

The authors claimed that the proposed deviated-reconstruction loss in Equation (8) could resolve the gradient problems, including both gradient obfuscation (L48, L72, etc.) and gradient vanishing/exploding (L10, L141, etc.). However, it is unclear how these problems are resolved by the proposed loss term.

This issue has three folds.

1. **[Minor] It is unclear what exact problem is being resolved** when the authors mixed the concepts of "gradient obfuscation" and "vanishing/exploding gradients." Specifically, "gradient obfuscation" is a general concept that also includes non-differentiable defense components, where the gradient is undefined and hence zero, yet "vanishing/exploding gradients" specifically refers to the defined but overly small or large gradients in differentiable defense components. While these are minor conceptual confusion, it is suggested to be consistent in the wording and clarify if it is the "non-differentiable" or "differentiable but vanishing/exploding" problem that the technique aims to solve, as their solutions typically require different strategies.
2. **[Moderate] It is unclear if the gradient issue indeed exists in previous evaluations.** While it is true that deep networks could have the problem of vanishing or exploding gradients, they have not been explicitly observed by previous evaluations of purification-based defenses, such as DiffPure [29]. Given this, it is hard to justify if the proposed techniques could obtain stronger results because they have indeed resolved the gradient issues or some other undiscovered reasons.
3. **[Major] It is unclear why adding a loss term B would resolve the gradient problem of the existing loss term A.** Specifically, if there were a gradient problem in the original robustness evaluation (with only $\mathcal{L}\_{cls}$), it means that the loss term $\mathcal{L}\_{cls}$ has already exhibited some form of gradient obfuscation (be it non-differentiable or vanishing gradients). In this case, why would adding a new loss term $\mathcal{L}\_{dev}$ would be able to resolve the gradient problem in $\mathcal{L}\_{cls}$? For example, if the total loss shows non-vanishing gradients, it is very likely that the gradient of $\mathcal{L}\_{cls}$ is still vanishing, and the observed gradient solely comes from the added $\mathcal{L}\_{dev}$. While it is explained how $\mathcal{L}\_{dev}$ can relieve the gradient problem (L153), it only explains how $\mathcal{L}\_{dev}$ relieve the gradient problem of itself but not $\mathcal{L}\_{cls}$.

**5. [Major] Unclear effectiveness of the proposed regularizer.**

Given that it is unclear how the gradient issue was resolved, I feel the effectiveness of $\mathcal{L}\_{dev}$ should require more exploration from two perspectives.
1. If it indeed solves the gradient problem of $\mathcal{L}\_{cls}$, the authors should provide evidence showing that the problem (1) exists before adding $\mathcal{L}\_{dev}$, and (2) disappears after adding $\mathcal{L}\_{dev}$.
2. If it cannot solve the gradient problem of $\mathcal{L}\_{cls}$ but mostly optimizes the gradient problem of itself, then it seems to me that making forward-backward samples deviated is more effective than increasing the classification loss, which is an interesting insight. In this case, I would recommend the authors explore the rationale behind this observation, and highlight the efforts of resolving gradient problems in $\mathcal{L}\_{dev}$ rather than $\mathcal{L}\_{cls}$. Additionally, an ablation study of the loss weight $\lambda$ should be included.

**6. [Moderate] Unclear discussion about randomized purification.**

The discussion at L234 is unclear (even with L674-676).
* First of all, if a single prediction has a high variance, the ultimate prediction should be an ensemble of multiple predictions, e.g., through majority vote. After that, it seems reasonable for attacks not to evaluate misclassified samples. From my understanding, this choice is not to reduce the computational cost, but a fact that you do not need to attack it -- you already find an adversarial example with distance zero.
* I did not see the logical connection between L234-L237 and L237-L240, mostly because it is unclear if the first part refers to the fixed clean image or the sampled images during the attack.
* Also, the effectiveness of this choice was not evaluated. Is this choice a must-have for the final results?

**7. [Minor] Visualization of purified noise.**

Since this paper was able to circumvent purification defenses, it is suggested to also visualize the adversarial noise before and after the purification defense. Right now only the pre-purification adversarial examples are provided.

### Clarity

**8. [Major] The presentation is confusing and contrived.**

* Be consistent when describing the gradient problem (detailed in Quality-4.1). The current presentation uses "gradient obfuscation" and "vanishing/exploding gradients" interchangeably, but the previous term is a general concept that also includes non-differentiable problems. In Section 3.2, jumping between these two terms and the proposed technique is very distracting. It is suggested to use only one of them and be clear about what specific problem you are trying to solve.
* The intuition behind Equation (8) is not clear until paragraph L166. It is suggested to explain the intuition (deviation, not the gradient problem) first.
* The paragraph L157 comes out of the blue and is very dense, it might fit more at the end of Section 3.2.
* Theorem 1 also comes out of the blue. The use of this theorem is unclear until L189. It is suggested to explain the remark (L187) or at least its high-level idea before introducing the theorem.
* The use of Theorem 3 was presented without any discussion.
* It is a bit hard to determine what exact defense was evaluated, it would be good to specify the defense name rather than citations.

**9. [Minor] Typos and notations.**

* Undefined terms
	* L151, $d()$
	* L172, "data density estimation"
	* L177, VP-SDSE
	* L206, $f\_d$ and $f\_r$
 * Typos
	* L142, Path -> Pass
	* L152, sampled -> reconstructed.

### Significance

N/A

**Questions:**

I am willing to raise my score if the following questions are adequately answered or clarified, ordered by importance.

1. Quality 4, 5
2. Quality 3
3. Quality 6 (merge to Quality 3 if they are relevant)
4. Originality 1
5. Quality 7
6. Fix Clarity 8

---

> ### Author Rebuttal · Authors · 2023-08-09
>
> We thank the reviewer for the valuable suggestions, and we are glad that the reviewer finds our work novel, timely, with strong results.
> > Q1: Related work for memory-efficient back-propagation
>
> Thanks for the suggestion. Prior works [a,b] propose the technique of gradient checkpointing for backpropagation with memory efficiency. They store fewer activations in forwarding passes and construct local computation graphs via recomputation. However, we are the first to apply a memory-efficient technique to attack purification defenses and resolve the problem of memory cost during attacks, which is a challenging problem as pointed out by [c,d]. We will add these detailed related work discussions in our revision.
> > Q2:  Attack framework vs. attack techniques
>
> Thanks for the suggestion. We will update our claim to make it clear as an attack technique.
> > Q3: Clarification of unbounded randomness
>
> Thanks for the great question. We clarify that DiffAttack tackles the randomness problem from two perspectives: 1) sampling multiple times in Eq. (8), and 2) optimizing perturbations for all samples including misclassified ones in all steps. Perspective 1) provides a more accurate estimation of gradients. Perspective 2) ensures a more effective attack optimization since the correctness of classification is of high variance over different steps. For instance, the classification result of a sample can be viewed as a Bernoulli distribution. We should reduce the success rate of Bernoulli distribution of samples by optimizing them with a larger attack loss, which would lead to a lower robust accuracy. In other words, one observation of failure in classification does not indicate that the sample has a low success rate statistically, and thus, 2) helps to continue optimizing the perturbations towards a lower success rate (i.e., away from decision boundary).
> > Q4 1): Gradient obfuscation vs. vanishing/exploding gradients
>
> Thanks for pointing it out. As gradient obfuscation is a more general concept, we will follow the suggestion and keep using vanishing/exploding gradients in our revision.
> > Q4 2): Gradient issues in prior evaluations
>
> Diffusion purification process induces an extremely deep graph. Specifically, DiffPure applies $100$ iterations of sampling and uses deep UNet with tens of layers as score estimators. Thus, the computational graph consists of thousands of layers, which could cause the problem of gradient vanishing/exploding. Prior work [d] also mentions the gradient problem (Section 4, 5.1 in [d]) and attempts to bypass it by BPDA attack, which is shown to be less effective compared with DiffAttack in Table 4. We will make it more clear.
> > Q4 3): Clarification of how added loss resolves gradient problem
>
> Thanks for the question. We propose the deviated-reconstruction loss, which is applied at the intermediate time steps of the computational graph, while the standard classification loss can only be applied at the end of the graph. Therefore, our proposed $L _{dev}$ will suffer less from the gradient problem compared to $L _{cls}$. We will clarify that *adding $L _{dev}$ will not solve the gradient problem of loss $L _{cls}$*. However, our $L _{dev}$ can lead to a deviated reconstruction of the image, which is easier to be misclassified and thus guides the optimization of searching adversarial examples effectively.
> > Q5: More explanations of the added regularizer
>
> Thanks for the feedback. $L _{dev}$ suffers less from the gradient problem compared with $L _{cls}$ (illustrated in Q4 (3)), and thus the objective of $L _{dev}$ can be optimized more easily, but it does not solve the gradient problem of $L _{cls}$. On the other hand, the optimization of $L _{dev}$ benefits the optimization of $L _{cls}$ in the way that $L _{dev}$ can induce a deviated reconstruction of the image with a larger probability of misclassification. We follow the suggestion to add ablation studies of the weights $\lambda$ to show the interplay of $L _{dev}$ and $L _{cls}$. A small $\lambda$ can weaken the deviated-reconstruction object and makes the attack suffer more from the vanishing/exploded gradient problem, while a large $\lambda$ can downplay the guidance of the classification loss and confuse the direction towards the decision boundary of the classifier. The results in Table 8 in the rebuttal PDF show that selecting $\lambda$ as 1.0 achieves better tradeoffs empirically, so we fix it as 1.0 for experiments.
> > Q6: Discussion about randomized purification
>
> Thanks for the question. As discussed in Q3, we can view classification results in steps as a Bernoulli distribution and optimize all samples towards a lower attack success rate. We agree that doing multiple evaluations and making optimization decisions can be an alternative to tackle the randomness problem with an accurate estimate of success rate.
> In L234-237, we point out the randomness problem induced by only optimizing the fixed clean images, and then we illustrate the proposed optimization technique in DiffAttack in L237-240. We will add these explanations for better clarification.
> > Q7: Visualization of samples after purification
>
> Following the suggestion, we add additional visualization in Figure 9 in the rebuttal PDF, showing that adversarial examples before and after the purification are stealthy.
>
> > Q8: Presentation and notations
>
> Thank you for the feedback. We will consistently use vanishing/exploded gradients. We illustrate the intuition of Eq. 8 (L166-173) before discussions in L148. We add high-level illustrations before Thm. 1 and 3. We also keep using names for all defenses.
>
> *[a] Chang et al. Reversible architectures for arbitrarily deep residual neural networks. AAAI 2018.*
>
> *[b] Gomez et al. The reversible residual network: Backpropagation without storing activations. NIPS 2017.*
>
> *[c] Nie et al. Diffusion Models for Adversarial Purification. ICML 2022.*
>
> *[d] Yoon et al. Adversarial purification with score-based generative models. ICML 2021.*

---

> > ### Comment · Reviewer_yE5n · 2023-08-11
> >
> > Thanks for the response and additional experiments. I have raised my score to 5 with the following two concerns.
> >
> > **Q4 & Q5: Regarding how the added loss term resolves the gradient issue.**
> >
> > I generally agree with the response regarding how the added loss term suffers less from the gradient problem. This conclusion, however, points out the non-usefulness of the original classification loss, which is an important observation (to me) but not addressed in the paper --- It seems that simply making the reconstructed image more deviated would suffice to reduce the defense's robustness, as least from the current presentation that there is a gradient problem with the classification loss (which is not explicitly demonstrated tho) and this paper does not resolve that directly.
> >
> > I have no problem with the proposed loss, but I feel it is important to make the above point clear in the paper. For example, would you be able to obtain a similar success rate if the classification loss was removed completely? If this is demonstrated (either true or false), the overall clarity (and potential contribution if demonstrated true) of this paper would be enhanced a lot. Beyond this, I am fine as long as the revised paper clarifies that the proposed loss does not resolve the gradient problem in the classification loss directly, but somehow circumvents that problem through a proxy. As a side suggestion, a more comprehensive evaluation could be showing the curves for the two loss terms, so readers would understand what is going on with the two losses.
> >
> > **Q3 & Q6: Regarding the unbounded randomness.**
> >
> > I agree with the two perspectives clarified in the rebuttal, which I already noted in my initial Q3. I think a better way to put my concern is that the current argument (for its novelty or importance) is a bit weak based on the following two points.
> >
> > First, why the randomness is "unbounded?" Previous defenses like DiffPure were also evaluated with EOT [A], which has the same spirit as Equation (8) in this paper, so I cannot see evidence that their randomness is not bounded yet this paper is bounded. This leads to the second point -- how Equation (8) is different from previous evaluations (in terms of resolving randomness)? I feel the root improvement is more relevant to the gradient problem of the original classification loss (rather than the design choices for resolving randomness). For example, it might be the case that EOT + the proposed loss is more useful than EOT + the original classification loss.
> >
> > Given the above two points, I feel the significance of resolving randomness can be tuned down in the paper (if I understand correctly). The most interesting part is the technical details of (1) how you apply EOT to the new proposed loss, and (2) what space you apply EOT. I think separating these discussions from the two main techniques could make the contributions more clear. Right now the randomness is presented as a novel problem (which is not the case as discussed above), yet the corresponding designs are not that significant (to me, given the previous application of EOT). I believe tuning down this part would not hurt the overall contribution a lot, since the other two parts are strong enough.
> >
> > [A] Synthesizing Robust Adversarial Examples. ICML 2018.

---

> > > ### Author Response · Authors · 2023-08-13
> > > **Follow-up Discussions with Reviewer yE5n [1/2]**
> > >
> > > > Q1: Regarding how the added loss term resolves the gradient issue.
> > >
> > > Thank you for the valuable feedback and suggestions! We add additional evaluations following the suggestion to explore whether the deviated-reconstruction loss suffices to reduce the defense’s robustness. We perform $\ell_\infty$ attacks against score-based purification with pretrained classifier WideResNet-28-10 on CIFAR-10. The results in Table 1 demonstrate that **only the deviated-reconstruction loss is not sufficient to attack the diffusion purification defenses**.
> > > We will clarify that although the classification loss suffers more from the gradient issue, it provides the necessary guidance to optimize the sample across the decision boundary.
> > >
> > > In the revised version, we will clarify that *the deviated-reconstruction loss does not resolve the gradient problem of the classification loss directly but circumvents the problem by leading to a deviated reconstruction of the image, which is easier to be misclassified and thus guides the optimization of searching adversarial examples effectively*. We will add the detailed explanations and discussion of the functionality of the deviated-reconstruction loss in responses to Q4 & Q5 in our rebuttal following the suggestion to make it clear. Thanks for the suggestions!
> > >
> > > Table 1: Attack performance (Robust accuracy (%)) of applying only deviated reconstruction loss (+$\ell_{dev}$), only classification loss (+$\ell_{cls}$), and the combinations of them (+$\ell_{cls}$+$\ell_{dev}$) against score-based purification with pretrained classifier WideResNet-28-10 on CIFAR-10 with different perturbation budgets $\epsilon$.
> > >
> > > | | +$\ell_{dev}$ | +$\ell_{cls}$ | +$\ell_{cls}$+$\ell_{dev}$ |
> > > | -------------------------------------------------- | ------------------ | ---------------- | ---------------- |
> > > |$\epsilon=8/255$ | 78.13 | 70.64 | 46.88 |
> > > | $\epsilon=4/255$ | 85.16 | 82.81 | 71.88 |
> > >
> > > We also evaluate the curves for the two loss terms (i.e., deviated-reconstruction loss $\ell_{dev}$ and cross-entropy loss $\ell_{cls}$) with score-based purification with WideResNet-28-10 on CIFAR-10 under $\ell_\infty$ DiffAttack with $\epsilon=8/255$. Comparing the loss curve of $\ell_{cls}$ without applying $\ell_{dev}$ (in Table 2) and the loss curve of $\ell_{cls}$ with applying $\ell_{dev}$ (in Table 3), we can see that the introduction of $\ell_{dev}$ guides the optimization of $\ell_{cls}$ effectively. We will make the results more clear by presenting them in figures in the revised version.
> > >
> > > Table 2: Loss curves of the cross-entropy loss $\ell_{cls}$ by applying only the classification loss $\ell_{cls}$ to attacking score-based purification with WideResNet-28-10 on CIFAR-10 under $\ell_\infty$ DiffAttack with $\epsilon=8/255$.
> > >
> > > | Epoch | 0 | 5 | 10 | 15 | 20 | 25 | 30 | 100 |
> > > | - | - | - | - | - | - | - | - | - |
> > > | $\ell_{cls}$ | 0.3856 | 0.6384 | 0.7283 | 0.7728 | 0.8028 | 0.8092 | 0.8091 | 0.8293 |
> > >
> > >
> > > Table 3: Loss curves of deviated-reconstruction loss $\ell_{dev}$ and cross-entropy loss $\ell_{cls}$ by applying $\ell_{dev}$ and $\ell_{cls}$ to attacking score-based purification with WideResNet-28-10 on CIFAR-10 under $\ell_\infty$ DiffAttack with $\epsilon=8/255$.
> > >
> > > | Epoch | 0 | 5 | 10 | 15 | 20 | 25 | 30 | 100 |
> > > | - | - | - | - | - | - | - | - | - |
> > > | $\ell_{cls}$ | 0.3856 | 0.8237 | 1.0289 | 1.1782 | 1.2843 | 1.3582 | 1.3726 | 1.3832 |
> > > | $\ell_{dev}$ | 0.2685	 | 0.6823 | 0.8394 | 0.9523 | 1.0383 | 1.1353 | 1.1534 | 1.1634 |

---

> > > ### Author Response · Authors · 2023-08-13
> > > **Follow-up Discussions with Reviewer yE5n [2/2]**
> > >
> > > > Q2: Regarding the unbounded randomness.
> > >
> > > Thank you for the valuable suggestions! We will follow the reviewer’s suggestions and tune down the significance of solving randomness and provide more details of the optimization process and analysis. We will illustrate how DiffAttack benefits resolving the randomness problem in detail as follows.
> > >
> > > First, we will point out that attacking against diffusion purification defenses suffers from the problem of large randomness because the diffusion and sampling process introduces large randomness which makes the calculated gradients unstable and noisy. Specifically, the randomness comes from the addition of Gaussian noises at each step in the diffusion process and sampling process. The randomness is large since there are typically multiple steps of Gaussian noise addition (e.g., 100 or more for score-based diffusion purification in DiffPure). We will rephrase “the problem of unbounded randomness’’ as “the problem of large randomness’’ in the abstract to avoid confusion.
> > >
> > > Second, we will clarify the strategy of approximating the expectation in Equation (8) by sampling multiple times and taking average benefits resolving the randomness problem similarly to EOT [A]. They both reduce the approximation error of the mean of random gradients by increasing the sample sizes. However, the benefits of the deviated-reconstruction loss in terms of the randomness challenge lie in the following perspective. When we expand the purification process as the computation graph, there is more randomness introduced at deep layers compared with shallow layers because more Gaussian noises are added at deeper layers. The deviated-reconstruction loss is applied uniformly across different layers of the computation graph and there is less randomness at shallow layers, while the classification loss is only added at the deepest layer of the graph and there is more randomness introduced at deep layers. Therefore, the deviated-reconstruction loss suffers less from the randomness problem, and the deviated-reconstruction loss with multiple samplings (EOT) outperforms the classification loss with multiple samplings (EOT).
> > >
> > > We will also include more technical details of applying EOT to the deviated-reconstruction loss following the suggestion. In particular, for a given batch of data, we can perform the diffusion process and sampling process and compute the gradient of the deviated-reconstruction loss with respect to the input data. We repeat the process multiple times and compute the average of the gradient across multiple times to estimate the mean of the gradient of deviated-reconstruction loss. We will follow the reviewer’s suggestions to add these discussions in the revision and update our claims to make the contributions more clear.
> > >
> > > *[A] Synthesizing Robust Adversarial Examples. ICML 2018.*

---

> > > > ### Comment · Reviewer_yE5n · 2023-08-13
> > > >
> > > > Thanks for the response. The new results look promising and insightful. I have raised my score to 6 accordingly.

---

> > > > > ### Author Response · Authors · 2023-08-13
> > > > > **Thank you for the valuable feedback**
> > > > >
> > > > > Thank you for the valuable feedback to help improve our work. We will definitely add the related discussions and analysis in our revision, and please let us know if you have further questions or suggestions!

---

### Official Review · Reviewer_EofG · 2023-07-05

**Soundness:** 4 excellent
**Presentation:** 3 good
**Contribution:** 3 good
**Rating:** 6
**Confidence:** 3

**Summary:**

This paper proposes a framework, DiffAttack, to attack against diffusion-based adversarial purification defenses. To address the issue of gradient vanishing/exploding, the paper introduces the concept of deviated-reconstruction loss and theoretically analyzes its relationship with density gradient estimation. To overcome the challenge of high memory cost, the paper presents a segment-wise forwarding-backwarding algorithm. Empirical experiments demonstrate the effectiveness of the proposed method and provide observations that contribute to a better understanding of the properties of the diffusion process.

**Strengths:**

1. The content is rich.
2. The theoretical analysis is solid.
3. The experimental results show improvements compared to the competing algorithm.


**Weaknesses:**

1. Section 3.2 discusses that maximizing the deviated-reconstruction loss in Equation (8) can lead to inaccurate data density estimation, which results in a sampling distribution that is inconsistent with the clean training distribution. However, it does not provide a detailed explanation of how this helps alleviate gradient vanishing/exploding.
2. Additional experiments should be conducted to investigate the transferability of DiffAttack.


**Questions:**

1. Line 151 does not provide a specific equation for α(·).
2. The paragraph from lines 157-165 should be placed before the paragraph from lines 147-156.

---

> ### Author Rebuttal · Authors · 2023-08-09
>
> > Q1: Detailed explanation of the benefits of deviated-reconstruction loss to alleviate gradient vanishing/exploding.
>
> Thanks for the suggestion. We will add the following explanations in Section 3.2. The deviate-reconstruction loss is applied at the intermediate time steps, and thus, the regularization is applied at relatively shallow layers of the computational graph compared to the standard classification loss, which is added at the end layer of the graph. The loss at relatively shallow layers has a shortened path of gradient backpropagation, thus relieving the gradient vanishing/exploding problem compared to the case only with the classification loss.
>
> > Q2: Additional experiments of transferability of DiffAttack.
>
> ACA [a] and Diff-PGD attack [b] explores the transferability of unrestricted adversarial attack, which generates realistic adversarial examples to fool the classifier and maintain the photorealism. They demonstrate that this kind of semantic attack transfers well to other models. To explore the transferability of adversarial examples by $\ell_p$-norm-based DiffAttack, we evaluate the adversarial examples generated on score-based purification withResNet-50 on defenses with pretrained WRN-50-2 and DeiT-S. The results in Table 9 in the one-page rebuttal PDF indicate that DiffAttack also transfers better than AdjAttack and achieves much lower robust accuracy on other models.
>
> *[a] Chen, Z., Li, B., Wu, S., Jiang, K., Ding, S., & Zhang, W. (2023). Content-based Unrestricted Adversarial Attack. arXiv preprint arXiv:2305.10665.*
>
> *[b] Xue, H., Araujo, A., Hu, B. and Chen, Y., (2023). Diffusion-Based Adversarial Sample Generation for Improved Stealthiness and Controllability. arXiv preprint arXiv:2305.16494.*
>
>
> > Q3: Specifications for $\alpha(\cdot)$.
>
> $\alpha(\cdot)$ is a time-dependent coefficient to regularize different time steps with different importance factors. We perform related ablation studies in Section 4.7 and observe that adding deviated reconstruction loss to uniformly sampled time steps achieves the best attack performance and tradeoff compared with that of adding loss to the same number of partial time steps only at initial/final stages. Specifically, the uniform sampling strategy corresponds to the formulation $\alpha(t)=1/k$ for $t ~ \text{mod} ~ (T/k) =0$, where $k \in Z^+$ controls the interval of uniform sampling. We will formulate the uniform sampling strategy explicitly in our revision.
>
> > Q4: Rearrangement of the explanations in line 157-165.
>
> Thanks for the suggestion. We rearrange the explanations in line 157-165 and the explanations of equation 8 in Section 3.2 for better coherency.

---

> > ### Comment · Reviewer_EofG · 2023-08-13
> > **Thanks for the rebuttal**
> >
> > Thanks for providing the rebuttal. I will keep my original rating and recommend acceptance.

---

### Official Review · Reviewer_3df8 · 2023-07-11

**Soundness:** 3 good
**Presentation:** 3 good
**Contribution:** 2 fair
**Rating:** 5
**Confidence:** 4

**Summary:**

The paper introduces an adversarial attack focused on diffusion-based purification techniques. In diffusion-based purification, a (malicious) input is first noised and then denoised by a diffusion model before being fed through the classifier. The paper aims at solving the issues that arise when using backpropagation-based PGD attacks due to the size of the denoising diffusion graph. The two main contributions are:

 - A per-timestep deviated reconstruction loss that maximizes the distance between the noisy version of the original image at a certain timestep and the denoising reconstruction.
 - Segment-wise forwarding-backwarding which is essentially gradient checkpointing for diffusion graphs

In the evaluation. they test their method against a variety of SOTA baselines on CIFAR10 and ImageNet and are able to beat previously best achieved robust accuracies by a margin.




**Strengths:**

 - The evaluation protocol seems reasonable to me and the results are good across the board compared to the given baselines.
 - Overall the presentation is clear
 - Both gradient checkpointing and the deviated reconstruction loss are quite simple ideas that seem to work well at solving the problem at hand.

**Weaknesses:**

 - I think the formulation of the segment-wise forwarding-backwarding algorithm makes it seem more novel than it truly is. Gradient checkpointing has been used to compute gradients in deep computational graphs by trading memory for compute in a lot of papers. I agree that diffusion graphs are well suited for this approach as the diffusion latents that are stored are very compact in size compared to the activations inside the score-net, but I still believe that it would be clearer to the reader to just refer to it as gradient-checkpointing.
 - I personally find some arrows in Figure 1 to be a bit confusing. I do not really understand the red arrows DiffAttack in the top part of the graph at t=0 and it's not super clear how they interact with the grey arrows for segment-wise forwarding-backwarding. I understand how the method works from the description in the text and I think it's a good idea to have a figure illustrating gradient flow in the paper but I believe clarity could be improved here.
 - In my opinion, Section 3 should contain a brief description of diffusion-based purification defenses in mathematical notation as the paper is explicitly aimed at attacking them.
 - I think it should be clearly stated at (11) that the goal is to maximize the objective similar to (6)/(7). Maybe it would even improve clarity to add a max close to (8) as minimization is more common in deep learning literature
 - I like Section 4.7 but I think it could be improved by showing results without the added deviated-reconstruction loss at all.
 - I believe the theoretical connection between Theorem 1 and the rest of the paper is rather weak. The Theorem only connects total variation to a mismatch between the ground-truth score function and the network estimate of the score. I think this would be a lot more useful if the paper would actually maximize the total-variation distance or if the Theorem could be stated in terms of the quantity being maximized.

**Questions:**

 - Especially in classifier guidance, it is common to denoise the current latent x_t to get a coarse estimate of x_0 and then calculate the gradient through this single-step approximation (For example "UPainting: Unified Text-to-Image Diffusion Generation with Cross-modal Guidance" Li et Al 2022 ). You mention several strategies in 3.2 to prevent gradient obfuscation by introducing additional losses at intermediate steps. Would it be possible to apply such an approach to your problem and do you expect it to work better/worse than your deviated-reconstruction loss?

**Limitations:**

 - I think the paper is very applicable to diffusion-based purifications and the authors do not mention strong limitations either.

---

> ### Author Rebuttal · Authors · 2023-08-09
>
> > Q1: Related work of the gradient checkpointing technique.
>
> Thanks for the suggestion. We will add discussions of the related work in our revision. To the best of our knowledge, prior works [a,b,c] propose the technique of gradient checkpointing to perform gradient backpropagation with memory efficiency. They store fewer activations during forwarding passes and construct the local computation graph via recomputation. However, we are the first to apply the memory-efficient backpropagation technique to attack diffusion purification defenses and resolve the problem of memory cost during attacks, which is realized as a challenging problem by prior attacks against purification defenses [d,e].
>
> *[a] Chen, Tianqi, Bing Xu, Chiyuan Zhang, and Carlos Guestrin. "Training deep nets with sublinear memory cost." arXiv preprint arXiv:1604.06174 (2016).*
>
> *[b] Chang, Bo, Lili Meng, Eldad Haber, Lars Ruthotto, David Begert, and Elliot Holtham. "Reversible architectures for arbitrarily deep residual neural networks." In Proceedings of the AAAI conference on artificial intelligence, vol. 32, no. 1. 2018.*
>
> *[c] Gomez, Aidan N., Mengye Ren, Raquel Urtasun, and Roger B. Grosse. "The reversible residual network: Backpropagation without storing activations." Advances in neural information processing systems 30 (2017).*
>
> *[d] Nie, Weili, Brandon Guo, Yujia Huang, Chaowei Xiao, Arash Vahdat and Anima Anandkumar. “Diffusion Models for Adversarial Purification.” International Conference on Machine Learning (2022).*
>
> *[e] Yoon, Jongmin, Sung Ju Hwang, and Juho Lee. "Adversarial purification with score-based generative models." In International Conference on Machine Learning, pp. 12062-12072. PMLR, 2021.*
>
> > Q2: Improvements of presentations in Figure 1.
>
> We apologize for the confusion in the figure. The red arrow in the top part denotes that the second line of examples is generated by DiffAttack, while the black arrow refers to the examples by standard adversarial (adv.) attacks. We will use another type of arrow to differentiate them from the arrows used for the classification flow. We compute the gradients with respect to the sample right before the classifier using standard techniques, and then the gradient backpropagation of DiffAttack is performed using a segment-wise forwarding-backwarding manner (the bidirectional grey dash line).
>
> > Q3: Mathematical formulation of diffusion purification defenses.
>
> Thanks for the suggestion. We will include the formulation of DDPM purification process (equation 1) and that of the score-based purification process (equation 3,4) with equation 7 for better understanding from a mathematical perspective.
>
> > Q4: Formulations related to optimization.
>
> Thanks for the suggestion. We emphasize that we maximize the loss objective in equation 11. We also add a symbol of $\max$ before equation 8 to clarify that we actually maximize the objective during DiffAttack as the following: $\max \mathcal{L}_{dev} = \mathbb{E}_t [\alpha(t) \mathbb{E} _{\mathbf{x}_t,\mathbf{x}'_t|\mathbf{x}_0} d(\mathbf{x}_t, \mathbf{x}'_t)]$.
>
> > Q5: Presentation of the results in Section 4.7.
>
> Thanks for the suggestion. We also add the result without deviated-reconstruction loss and provide the results in Table 10 in the one-page rebuttal PDF.
>
> > Q6: Clarification of the connection between Theorem 1 and the rest of the paper.
>
> Theorem 1 indicates that deviated-reconstruction loss in Equation (8) induces inaccurate data density estimation, which results in the discrepancy between the sampled distribution and the clean training distribution. Therefore, it demonstrates that the deviated-reconstruction loss can induce inaccurate reconstruction of the clean distribution and benefit the optimization of adversarial examples. In the analysis, the total variation distance is used to quantify the distribution distance for rigorous derivation. In practice, we cannot compute the total variation distance between complex data distributions and can only optimize via finite samples. Therefore, we design the deviated-reconstruction loss in Equation 8 for the ease of optimization on sampled instances.
>
> > Q7: Discussion of the application of another strategy in the setting.
>
> UPainting [f] adds image-text matching guidance at the intermediate steps of the sampling process to promote the effectiveness of conditional scene generations. The main difference with DiffAttack is that UPainting attempts to promote cross-modal alignment while DiffAttack attempts to relieve the problem of gradient obfuscation via intermediate guidance. The idea of providing classification-guided regularization based on a one-step approximation of the clean image looks promising. However, after some initial trials of evaluations, we found that the accuracy of one-step denoised sample is pretty low for samples at the initial stage. Specifically, the accuracy is below $30\\%$ for half of the time steps on CIFAR-10 with WideResNet-28-10. Therefore, it is a concern that the classification guidance would not be effective. It would be interesting for future work to use more sophisticated methods to provide more effective classification guidance. For example, training a classifier with small Gaussian augmentations would possibly benefit a good intermediate classifier since the one-step approximation of the clean image is clearly inaccurate and mixed with Gaussian noises.
>
> *[f] "UPainting: Unified Text-to-Image Diffusion Generation with Cross-modal Guidance" Li et al 2022.*

---

> > ### Comment · Reviewer_3df8 · 2023-08-17
> >
> > I would like to start by thanking the authors for the rebuttal.
> >
> > - Q1: I do agree that previous works focus on "inner network checkpointing" whereas this method checkpoints between the solver steps. I believe it's a  valid contribution. I just think it would be easier to understand the idea for people that know checkpointing by shortly referencing it.
> >  - Q2  - Q4: Thank you for these updates and improving clarity!
> >  -  Q5: Thank you for the experiments. These results show that the additional regularization term helps a lot with the problem at hand.
> >  -  Q6: I still believe that there is a certain gap between the theoretical results and the practical formulation but I understand that it is not possible to actually calculate the TV distance. Theoretical insights that directly analyze the connection with the quantity that is optimized would be ideal though.
> >
> > Overall, the paper shows some strong results on a relevant problem. The method itself is not super novel but effective and I believe the paper is worth acceptance if some of the clarity issues from Q2-Q4 are resolved.

---

> > > ### Author Response · Authors · 2023-08-18
> > > **Follow-up Discussion with Reviewer 3df8**
> > >
> > > Thank you for the valuable feedback and suggestions!
> > >
> > > We will reference the gradient checkpointing technique when introducing the segmentwise forwarding-backwarding algorithm in Section 3.3 for better understanding and also incorporate the related work of gradient checkpointing in Section 5 for completeness. Thank you for the suggestions!
> > > We will definitely include all the modifications in Figure 1, equation 7, and equation 11 for better clarification in the revised version. We will also replace Table 5 in the manuscript with Table 10 in the one-page rebuttal PDF for a better presentation of the results.
> > >
> > > Thanks for the valuable comments about Theorem 1. We plan to further reduce the gap between the analysis and practical formulation following the suggestion as follows. In Theorem 1, $q \_t$ is a Gaussian distribution with the mean $\tilde{x} \_t$ and $q’ \_t$ is a Gaussian distribution with the mean $\tilde{x}’ \_t$.
> > > We analyze the objective $D_{TV}(q \_t,q’ \_t)$ in Theorem 1 and show that maximizing it can induce inaccurate data density estimation and thus inaccurate sample reconstruction. In the deviated-reconstruction loss, we maximize the vector distance between $\tilde{x} \_t$ and $\tilde{x}’ \_t$, which are the mean of Gaussian distribution $q \_t$ and $q’ \_t$, respectively. The connection of the two objectives can be built by providing more theoretical insights about why maximizing the distance of mean of Gaussians (i.e., $|\tilde{x} \_t - \tilde{x}’ \_t|$) can implicate maximizing the TV distance between Gaussians (i.e., $D_{TV}(q \_t,q’ \_t)$). Theorem 1.3 in [a] shows that for one dimensional Gaussian $\mathcal{N}(\mu \_1,\sigma \_1^2)$ and $\mathcal{N}(\mu \_2,\sigma \_2^2)$, the TV distance can be lower bounded by a function of the distance of mean values as the following:
> > > $$
> > > \dfrac{1}{200} \min \\{1, \max \\{ \dfrac{40|\mu \_1-\mu \_2|}{\sigma \_1}, \dfrac{|\sigma \_1^2-\sigma \_2^2|}{\sigma \_1^2} \\}  \\} \le D_{TV}(\mathcal{N}(\mu \_1,\sigma \_1^2),\mathcal{N}(\mu \_2,\sigma \_2^2)),
> > > $$
> > > from which we can see that maximizing the distance of the means (i.e., $|\mu \_1-\mu \_2|$) can indicate maximizing the TV distance $D_{TV}(\mathcal{N}(\mu \_1,\sigma \_1^2),\mathcal{N}(\mu \_2,\sigma \_2^2))$. Similar insights are shown for multi-dimensional Gaussians in Theorem 1.2 in [a].
> > > We will add the theoretical intuitions and related discussions in Theorem 1 in Section 3.2 for clarification following the suggestions.
> > >
> > > We are glad that the reviewer finds our method effective with strong results and worth acceptance. Please let us know if there are other comments and suggestions. Thank you for your valuable suggestions again!
> > >
> > > *[a] Devroye, Luc, Abbas Mehrabian, and Tommy Reddad. "The total variation distance between high-dimensional Gaussians with the same mean." arXiv preprint arXiv:1810.08693 (2018).*

---

### Author Rebuttal · Authors · 2023-08-09

We thank all the reviewers for their comments and valuable feedback. We are glad that the reviewers find our work novel, effective, and well-written. We have made the following major updates following the reviews’ suggestions to further improve our work.

1. We add related work of memory-efficient gradient back-propagation to the literature review, following the suggestions from Reviewer yE5n and 3df8.

2. We make more clarifications of the benefits of the deviated-reconstruction loss to the gradient problems, following the suggestions from Reviewer yE5n.

3. We provide more visualizations to demonstrate the stealthiness of DiffAttack, following the suggestions from Reviewer yE5n and Vu7G.

4. We add additional ablation studies of the loss weights, following the suggestions from Reviewer yE5n and Vu7G.

5. We explore the transferability of DiffAttack for different model architectures, following the suggestions from Reviewer Vu7G and EofG.

6. We improve the presentations of the arrows in Figure 1 and the mathematical formulation of the optimization problem, following the suggestions from Reviewer 3df8.

---

### Decision · Program_Chairs · 2023-09-21

**Decision:**

Accept (poster)

**Comment:**

This paper introduces an attack on diffusion based defenses to adversarial examples. The attack is effective and improves attack success rates by 10-20%. The reviewers raise a number of important points of clarification, but almost all of these were addressed to both the reviewers and my satisfaction during the discussion period. Ultimately after discussion all reviewers recommend acceptance and I agree here: the paper makes a valuable contribution. I encourage the authors to make the clarifications to the paper that helped the reviewers see what was being done.